# HelloBench: Evaluating Long Text Generation Capabilities of Large Language Models

## Abstract

In recent years, Large Language Models (LLMs) have demonstrated remarkable capabilities in various tasks (e.g., long-context understanding), and many benchmarks have been proposed. However, we observe that long text generation capabilities are not well investigated. Therefore, we introduce the Hierarchical Long Text Generation Benchmark (**HelloBench**), a comprehensive, in-the-wild, and open-ended benchmark to evaluate LLMs' performance in generating long text. Based on Bloom's Taxonomy, HelloBench categorizes long text generation tasks into five subtasks: open-ended QA, summarization, chat, text completion, and heuristic text generation. Besides, we propose Hierarchical Long Text Evaluation (**HelloEval**), a human-aligned evaluation method that significantly reduces the time and effort required for human evaluation while maintaining a high correlation with human evaluation. We have conducted extensive experiments across around 30 mainstream LLMs and observed that the current LLMs lack long text generation capabilities. Specifically, first, regardless of whether the instructions include explicit or implicit length constraints, we observe that most LLMs cannot generate text that is longer than 4000 words. Second, we observe that while some LLMs can generate longer text, many issues exist (e.g., severe repetition and quality degradation). Third, to demonstrate the effectiveness of HelloEval, we compare HelloEval with traditional metrics (e.g., ROUGE, BLEU, etc.) and LLM-as-a-Judge methods, which show that HelloEval has the highest correlation with human evaluation.

## 1 Introduction

In recent years, Large Language Models (LLMs) (Achiam et al., 2023; Touvron et al., 2023; Bai et al., 2023a) have demonstrated impressive performance across multiple natural language processing (NLP) tasks (e.g., Machine Translation, Sentiment Analysis, Dialogue System, etc.) (Yao et al., 2023; Zhang et al., 2023b; Yi et al., 2024). Besides, as the importance of the long-context capabilities of LLMs grows (Li et al., 2023), numerous evaluation benchmarks related to long-context (Li et al., 2024; Wang et al., 2024b; Zhang et al., 2024c) along with methods for improving the long-context capabilities of LLMs (Peng et al., 2023; Chen et al., 2023; Liu et al., 2024a) have emerged. Nevertheless, existing long-context research focuses on the capabilities of LLMs to understand, retrieve, and process long input text, with limited research (Köksal et al., 2023; Tan et al., 2024; Pham et al., 2024; Bai et al., 2024) concentrating on the long text generation capabilities. Besides, long text generation capabilities are essential for LLMs, as they meet the users' demands for long output text, such as long story writing (Xie & Riedl, 2024) and long essay writing. We can also see the importance of long text generation capabilities through the iterative updates of OpenAI's series of models. The maximum output tokens have increased from 4,096 in GPT-4o (OpenAI, 2024) to 16,384 in GPT-4o-2024-0806, and recently to 32,768 in the o1-preview and 65,536 in the o1-mini[1]. The strong reasoning capabilities of o1-mini and o1-preview are also related to their capabilities to generate long reasoning chains, which highlight the importance of long text generation capabilities.

However, there is a significant shortfall in a comprehensive benchmark for evaluating the capabilities of LLMs to generate long text. To mitigate this shortfall, there are two main issues to address: *how*

---

[1]https://openai.com/o1/

Figure 1: The overview of HelloBench. *(In the Middle)*: The six levels of Bloom's Taxonomy, from bottom to top, are **remember**, **understand**, **apply**, **analyze**, **evaluate**, and **create**. These correspond to the five tasks in HelloBench and HelloEval. Detailed examples are provided in Appendix A.

*to construct a comprehensive long text generation benchmark for LLMs?* and *how to evaluate the long text generation capabilities accurately with minimal human evaluation?*

Therefore, in this work, we introduce the Hierarchical Long Text Generation Benchmark (**HelloBench**), a comprehensive, in-the-wild, and open-ended benchmark to evaluate LLMs' capabilities to generate long text. As shown in Figure 1, based on Bloom's Taxonomy (Anderson & Krathwohl, 2001), the long text generation capabilities of LLMs are categorized into six hierarchical levels: remember, understand, apply, analyze, evaluate, and create. The levels of remembering, understanding, applying, analyzing, and creating correspond to specific tasks in HelloBench: **open-ended QA**, **summarization**, **chat**, **text completion**, and **heuristic text generation**, respectively. Specifically, to construct a high-quality HelloBench, we manually collected and filtered data from the internet and publicly available datasets to obtain the most natural data for long text generation tasks that are in-the-wild and open-ended. Finally, HelloBench includes 647 samples, covering 5 categories and 38 subcategories. The differences between HelloBench and the previous benchmarks are shown in Table 1.

Table 1: A comparison of our HelloBench with some notable datasets. "Comprehensive" means that the benchmark has more than 4 tasks or categories. "In-The-Wild" means that the benchmark is sourced from real user scenarios. "Open-Ended" means that the answers in the benchmark are not fixed, and the evaluation method does not rely on gold answers. "Long-Output" means that the benchmark requires LLMs to generate text at least 1,000 words.

| Benchmarks | Comprehensive | In-The-Wild | Open-Ended | Long-Output |
|---|---|---|---|---|
| LongForm-C (Köksal et al., 2023) | ✗ | ✗ | ✓ | ✓ |
| ELI5 (Fan et al., 2019) | ✗ | ✓ | ✗ | ✗ |
| Suri (Pham et al., 2024) | ✗ | ✗ | ✓ | ✓ |
| LongBench-Write (Bai et al., 2024) | ✗ | ✓ | ✓ | ✓ |
| ProxyQA (Tan et al., 2024) | ✗ | ✓ | ✓ | ✓ |
| LongBench (Bai et al., 2023b) | ✓ | ✗ | ✗ | ✗ |
| HelloBench (Ours) | ✓ | ✓ | ✓ | ✓ |

For the level of evaluating in Bloom's Taxonomy, we introduce a human-aligned evaluation method **HelloEval** to evaluate LLMs' long text generation capabilities using LLM-as-a-Judge (Zheng et al., 2024). Specifically, although the best approach for open-ended text evaluation is human evaluation (Chang et al., 2024), there are two drawbacks on human evaluation for long text generation. First, human evaluation is time-consuming and labor-intensive, especially when evaluating the quality of long text. Second, providing an overall evaluation score for a long text is challenging for humans due to the difficulties in understanding a long text and inherent subjective biases among humans. To address these issues, as shown in Figure 2, our proposed HelloEval aims to align with human evaluation with significantly reduced time and effort. Specifically, HelloEval includes two stages (i.e., the preparation stage and the execution stage), the first stage prepares the human annotation data on the checklists evaluation and overall score evaluation, and then we use linear regression to fit the weighted scores of checklists. In the second stage, we use LLM-as-a-Judge to evaluate the results of checklists, and then use weighted scores of checklists to get an overall score.

Based on HelloBench and HelloEval, in Table 2, we have evaluated long text generation capabilities on about 30 open-source and proprietary LLMs, as well as specialized LLMs for long text generation, and we have the following findings: (1) Current well-performed LLMs (e.g., GPT-4o (OpenAI, 2024), Claude-3.5-Sonnet (Antropic, 2024)) struggle to generate text longer than 4000 words, regardless of whether the instructions include explicit or implicit length constraints. Though they perform acceptably when generating short text, their output length remains quite limited, typically around 2,000 words. (2) Some open-source LLMs (e.g., LongWriter-GLM4-9B, Suri-I-ORPO) can generate long text, but the generated texts exhibit severe repetition and significant quality degradation. (3) We have compared the LLMs before and after enhancement in long-context capabilities, further observing that there exists a negative correlation between LLMs' long-context understanding capabilities and their long text generation capabilities. (4) HelloEval achieves the highest correlation with human evaluation compared to traditional metrics (e.g., ROUGE (Lin, 2004), BLEU (Papineni et al., 2002), PPL, etc.) and various LLM-as-a-Judge evaluation methods.

Our main contributions are as follows:

- We construct a comprehensive, in-the-wild, and open-ended benchmark to evaluate the long text generation capabilities of both open-source and proprietary LLMs.

- We propose a human-aligned evaluation method HelloEval to evaluate the long text generation capabilities of LLMs. Compared to traditional metrics and LLM-as-a-Judge evaluation methods, HelloEval achieves the highest correlation with human evaluation.

- We conduct comprehensive experiments to evaluate the long text generation capabilities of about 30 LLMs and provide detailed discussions on the limitations and future directions.

## 2 HELLOBENCH

### 2.1 OVERVIEW OF HELLOBENCH

To comprehensively and accurately evaluate the long text generation capabilities of LLMs, we adopt the concept of Bloom's Taxonomy (Anderson & Krathwohl, 2001) and classify the cognitive levels of LLMs into six hierarchical levels: remember, understand, apply, analyze, evaluate, and create. We build HelloBench based on these levels, and prepare corresponding tasks for each level as follows:

1. **Remember**: we use open-ended QA (Wang et al., 2024c) to represent the capabilities of LLMs to remember, as LLMs need memory to respond to open-ended questions.

2. **Understand**: we use summarization (Jin et al., 2024) task, especially long summarization task to represent the capabilities of LLMs to understand, as summarization requires fundamental understanding ability.

3. **Apply**: we use the chat (Zheng et al., 2023) task to represent the capabilities of LLMs to apply, as LLMs need to act as chatbots in real application scenarios.

4. **Analyze**: we use the text completion (Park & Park, 2020) task to represent the capabilities of LLMs to analyze, as analysis of the preceding text is necessary for a good completion.

5. **Evaluate**: we use the LLM-as-a-Judge (Zheng et al., 2024) in HelloEval to evaluate the long text generation capabilities.

6. **Create**: we use heuristic text generation task (Venkatraman et al., 2024) to represent the LLMs' capabilities to create, as LLMs need to generate text based on the heuristic prompts.

Some tasks may correspond to multiple hierarchical levels. However, we have selected representative tasks that frequently appear in real-world scenarios and mapped them to the cognitive level. During the construction of HelloBench, we adhere to four core requirements: (1). **Comprehensive**: To enhance the diversity of the dataset, the five tasks of HelloBench have subcategories detailed in Section 2.2. (2). **In-The-Wild**: We ensure that the data is based on real-world scenarios, so the evaluation remains practical. (3). **Open-Ended**: All data should be open-ended. Besides, we collecte the latest and most original data from real users, which guarantees that the data is not leaked to LLMs' pretraining stage. (4). **Long-Output**: We verify from both data sources and manual checks that each data requires a long output. Thus for each HelloBench data, LLMs implicitly generate long text.

## 2.2 DATASET COLLECTION

In this section, we briefly introduce the task definitions and the data collection approach for the tasks in HelloBench. Please refer to Appendix B for detailed information on dataset collection and prompt wrapping. Please refer to Appendix C for the data quality of HelloBench.

**Open-Ended QA**  Question Answering (**QA**) is a classic task for LLMs (Hendrycks et al., 2020). Currently, closed-ended QA, such as multiple-choice QA, is more commonly used. In long text generation, we focus on open-ended QA to evaluate the long text generation capabilities of LLMs because open-ended questions usually require more detailed and lengthy responses. We collected the latest 200 open-ended questions from Quora[2]. To be specific, we first collected around 40 questions from each of the 10 most popular topics, preferring questions that are more recent and have higher response activity. After manually filtering, we kept about 20 questions per topic.

**Summarization**  Summarizing long documents poses significant challenges for LLMs in both comprehension and generation (El-Kassas et al., 2021). Specifically, we collected samples from seven publicly available summarization datasets, where the source documents range from 3,000 to 6,000 words to make them suitable for long summarization tasks. After manually filtering, we excluded low-quality documents and obtained five distinct subcategories: news, blogs, academic articles, reports, and long dialogue summarization, where each subcategory includes 20 samples.

**Chat**  To evaluate LLMs' application capabilities to generate long text and understand its practical importance, we construct the chat tasks based on WildChat (Zhao et al., 2024). WildChat collected conversations between users and LLMs in real-world scenarios, we selected conversations where the model's responses were over 1,000 words. To ensure the diversity of the chat tasks and explore the distribution of long text generation scenarios in WildChat, we follow InsTag (Lu et al., 2023) to label conversations using GPT-4o and normalize these labels. After that, we obtained 15 subcategories with 147 samples and observed that over 10k conversations have responses with over 1,000 words.

**Text Completion**  Since LLMs produce sequential output, text completion is a natural task for evaluating LLMs' capabilities to generate long text (Kang & Hovy, 2020). Specifically, we pre-defined three text completion subcategories: continuation, imitation, and style transfer. In real scenarios, story-based text completion tasks are more natural and novel. Thus, the text completion tasks are story-based. To ensure the originality and timeliness of the stories, we collected around 200 stories from the subreddit r/shortstories[3], where users share and discuss original short stories in the wild. After manually filtering to retain high-quality and longer stories, we kept around 80 samples in total.

**Heuristic Text Generation**  Heuristic text generation is defined as creating content based on given heuristic writing prompts. We found that many users request LLMs to write a long story, essay, report, etc in WildChat. Thus, we pre-defined five heuristic text generation subcategories and collected data from various internet sources. After filtering, we kept around 20 samples for each subcategory.

## 2.3 DATASET STATISTICS

Figure 5 presents all the categories and subcategories in HelloBench along with their proportions, where more details are shown in Table 8. Figure 8 and Table 9 show the word lengths of instructions in HelloBench, where we use NLTK (Loper & Bird, 2002) to tokenize the sentences into words.

## 3 HELLOEVAL

### 3.1 PIPELINE OF HELLOEVAL

Evaluating long text generation is difficult for both humans and LLMs. To address this, we use checklists to break the evaluation into two steps. The first step evaluates checklist results, while the second step evaluates the overall score. Checklists enhance the interpretability and reliability

---

[2]https://www.quora.com/

[3]https://www.reddit.com/r/shortstories/

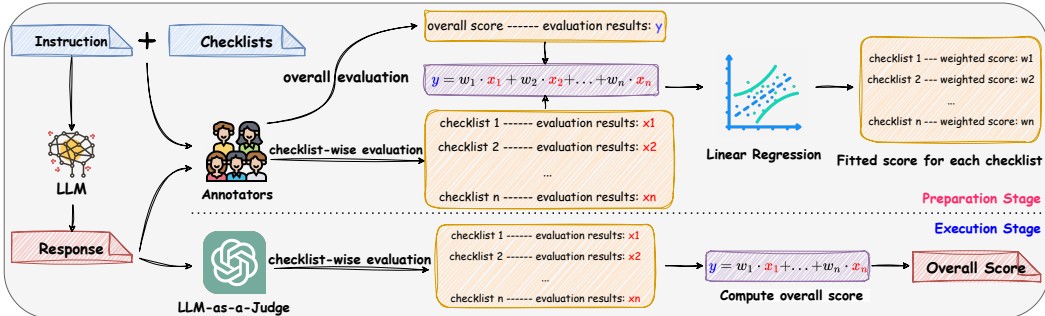

Figure 2: The pipeline of HelloEval has two stages. (*top*): In the preparation stage, we aim to determine the weighted score for each checklist. First, we have human annotators assign checklist results to each instruction-response pair. Then, the annotators give an overall score. By using linear regression, we can obtain the weighted scores for the checklists that align with humans. (*bottom*): In the execution stage, we use LLM to evaluate the checklist results for the instruction-response pairs, and then sum these scores based on the previously fitted weighted scores to get the overall score.

of final evaluations. For each instruction in HelloBench, the checklists consist of 4-6 yes or no questions to evaluate specific aspects of the response quality. Previous studies (Lin & Chen, 2023; Liu et al., 2023; Fu et al., 2023) often assign separate scores for different aspects of response quality, averaging them for the overall score. This method overlooks the varying impact of each checklist on the final score, which is crucial in open-ended text evaluation. Additionally, previous checklist-based approaches (Pereira & Lotufo, 2024; Lee et al., 2024; Lin et al., 2024) either use checklists as prefixes in prompt or average all checklist scores, fail to maximize the potential of the checklists and treat the influence of each checklist on the final score as equal, leading to significant evaluation bias. To address these issues, we propose HelloEval for evaluating long text generation tasks. As shown in Figure 2, HelloEval is divided into two stages. In the preparation stage, we carefully design checklists for each subcategory of HelloBench. We then collect (instruction, response, checklists) pairs from different LLMs. Annotators evaluate whether each checklist is satisfied based on the instruction and response, and also provide an overall score based on evaluation results of checklists. After collecting multiple data points, we use linear regression to fit the data and obtain a weighted score for each checklist, which enables the alignment of each checklist weighted score with human evaluation implicitly. In the execution stage, we use LLM-as-a-Judge (Zheng et al., 2024; Zhu et al., 2023). Given a long text and the associated checklists, LLMs can effectively evaluate the checklist results. Using the weighted scores fitted from the preparation stage, we can calculate the overall score for the response. The construction of checklists and the details of human annotation are provided in Appendix E and Appendix F.

## 3.2 REGRESSION ANALYSIS

To obtain the weighted scores for the checklists, we perform a linear regression analysis on the human annotation data, fitting the linear contribution of each checklist to the overall score and obtaining corresponding weighted scores. The linear regression formula is:

$$y = \sum_{i=0}^{n} w_i x_i = w_1 x_1 + w_2 x_2 + ... + w_n x_n, \tag{1}$$

where $y$ represents the overall score, while $x_1, x_2, \ldots, x_n$ are the evaluation results from each checklist, and $n$ is the number of checklits. The weights $w_1, w_2, \ldots, w_n$ are the values we need to fit for each checklist's contribution to the overall score. To ensure the robustness of the fitting results, we hire five annotators to annotate and we collect the human annotation data from LLaMA-3.1-8B (Meta, 2024), Qwen-2-7B (Yang et al., 2024), Claude-3.5-Sonnet, and GPT-4o-Mini, ensuring robustness of fitting results to different LLMs. The specific fitting results and fitting analysis are provided in Appendix G.

### 3.3 LLM-AS-A-JUDGE

LLM-as-a-Judge (Zheng et al., 2024; Chen et al., 2024) refers to using LLMs as evaluators to evaluate the capabilities of LLMs. Recently, this approach has been widely used to replace time-consuming and labor-intensive human evaluations, especially for open-ended text evaluation. In the context of long text generation, checklist-wise evaluation requires LLMs to answer 4-6 yes or no questions by reading a long text, similar to classic reading comprehension tasks (Xiao et al., 2023). Given that LLMs have strong reading comprehension capabilities, sometimes surpassing those of humans (OpenAI, 2024), we believe that using LLM-as-a-Judge for checklist-wise evaluation is feasible and reasonable. Specifically, we chose to have the LLM-as-a-Judge evaluate all checklists for a given instruction-response pair at once to save on resource consumption, rather than having the LLM-as-a-Judge evaluate one checklist at a time. For the choice of LLM-as-a-Judge, we selected GPT-4o, we also recommend using GPT-4o-Mini as LLM-as-a-Judge, which can save a lot of costs. The prompt template for checklist-wise evaluation is shown in Figure 9. To further demonstrate the reasons for choosing GPT-4o as the LLM-as-a-Judge and the effectiveness of the LLM-as-a-Judge, we have conducted experiments, which are provided in Appendix I.

## 4 EXPERIMENTS

### 4.1 EXPERIMENTAL SETUP

**Evaluated Models** In this work, we mainly evaluate 10 proprietary LLMs, 15 mainstream open-source LLMs, and 2 long text generation capabilities enhanced LLMs. All LLMs are chat or instruct versions. Detailed information is provided in Appendix J.1. For all LLMs, following (Song et al., 2024), we set a unified generation configuration for fair comparison: temperature is set to 0.8 and the max new tokens are set to 16,384 (if less than 16,384, set it to the maximum of the model). All experiments are done in the same computation environment with 8 NVIDIA 80GB A800 GPUs.

**Evaluation Metrics** We use the **"S" (Score)** as the overall score of the long text generated by LLMs, which is aligned with the human evaluation with the help of HelloEval as shown in Section 4.5. **"WC" (Word Count)** is an observation metric used to measure how many words LLMs can generate in long text tasks. The larger "S" shows higher long text generation qualities.

**Score Rescaling** To further clearly show the differences between various LLMs, we follow (Lin et al., 2024) to rescale the scores. Specifically, our checklist-wise evaluation has five grading levels: 0, 0.25, 0.5, 0.75, and 1, with 0.75 indicating an acceptable response. Therefore, the rescaling formula is $S = (score - 0.75) \times 4$. The range of scores has changed from $[0, 100]$ to $[-300, 100]$, where a positive score indicates that the LLM can generate acceptable long text.

### 4.2 MAIN EXPERIMENTS

We first evaluate the long text generation capabilities of 9 proprietary LLMs, 12 open-source LLMs, and 2 capability-enhanced LLMs on five tasks of HelloBench. In this part, instructions in HelloBench impose an implicit constraint on the output length of the LLMs, such as {The article should be long enough to thoroughly explore the topic}, without specifying an **exact word count constraint**. The experimental results are shown in Table 2. We summarize our findings as follows:

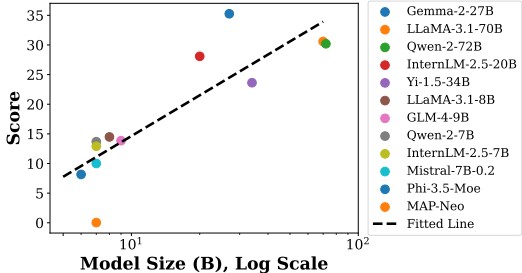

Figure 3: Scaling Law of Model Size and Performance for open-source LLMs.

(1) **Comparison of different LLMs.** Proprietary LLMs have average scores ranging from 24.78 (GLM-4-API) to 48.55 (GPT-4o-2024-08-06), while open-source LLMs have average scores ranging from 0.03 (MAP-Neo) to 35.26 (Gemma-2-27B). This indicates that proprietary LLMs have superior long text generation capabilities compared to open-source LLMs. Figure 3 shows the scores for different model sizes of LLMs, we find that larger models generally yield higher scores. Within the

Table 2: Main Experiments: The evaluation results of open-source LLMs, proprietary LLMs, and long text generation capabilities enhanced LLMs on HelloBench. "OEQA" represents open-ended QA, "Summ" represents summarization, "TC" represents text completion, "HTG" represents heuristic text generation, "AVG" represents average score on five tasks, "S" represents rescaled score, and "WC" represents word count. The results are in descending order.

| Models | OEQA | | Summ | | Chat | | TC | | HTG | | AVG | |
|---|---|---|---|---|---|---|---|---|---|---|---|---|
| | S | WC | S | WC | S | WC | S | WC | S | WC | S | WC |
| ⚔ Proprietary Large Language Models | | | | | | | | | | | | |
| GPT-4o-2024-08-06 | 54.82 | 898 | 29.71 | 457 | **42.88** | 1436 | **67.49** | 1581 | 47.87 | 1121 | **48.55** | 1098 |
| Mistral-Large-API | 53.15 | 728 | 34.04 | 652 | 32.62 | 1379 | 66.99 | 1350 | 47.07 | 859 | 46.77 | 994 |
| o1-Mini | 46.85 | 1858 | **38.57** | 813 | 38.75 | 2462 | 57.47 | 1762 | **48.75** | 1353 | 46.08 | 1650 |
| Claude-3.5-Sonnet | **62.73** | 750 | 31.34 | 388 | 32.60 | 1136 | 51.27 | 1068 | 40.92 | 941 | 43.77 | 857 |
| Gemini-1.5-Pro | 53.11 | 692 | 23.55 | 463 | 27.65 | 1381 | 44.29 | 921 | 47.59 | 783 | 39.24 | 848 |
| Deepseek-API | 44.31 | 801 | 18.50 | 424 | 33.04 | 1320 | 47.62 | 1441 | 34.97 | 754 | 35.69 | 948 |
| Yi-Large | 48.31 | 679 | 23.13 | 486 | 16.53 | 1190 | 45.78 | 1020 | 31.23 | 766 | 32.99 | 828 |
| Qwen-Max | 50.79 | 655 | 12.07 | 273 | -1.37 | 966 | 43.94 | 779 | 36.39 | 705 | 28.36 | 676 |
| GLM-4-API | 47.49 | 845 | 8.38 | 395 | 3.76 | 901 | 34.64 | 879 | 29.66 | 871 | 24.78 | 778 |
| 🛡 Open-Source Large Language Models | | | | | | | | | | | | |
| Gemma-2-27B | 52.38 | 680 | 17.78 | 381 | 18.10 | 1170 | 41.77 | 920 | 46.25 | 741 | 35.26 | 778 |
| LLaMA-3.1-70B | 48.13 | 867 | 20.66 | 611 | 26.99 | 1358 | 25.27 | 1466 | 31.84 | 910 | 30.58 | 1042 |
| Qwen-2-72B | 48.79 | 668 | 26.59 | 894 | 5.04 | 949 | 34.90 | 1657 | 35.66 | 740 | 30.20 | 982 |
| InternLM-2.5-20B | 51.27 | 740 | 8.65 | 324 | 5.81 | 1278 | 36.68 | 989 | 37.97 | 817 | 28.08 | 830 |
| Yi-1.5-34B | 47.36 | 751 | -14.33 | 328 | 5.02 | 1205 | 44.73 | 1054 | 35.31 | 875 | 23.63 | 843 |
| LLaMA-3.1-8B | 42.52 | 801 | 15.77 | 640 | -5.26 | 1450 | -5.61 | 3138 | 24.99 | 965 | 14.48 | 1399 |
| GLM-4-9B | 40.71 | 788 | -5.38 | 329 | 0.47 | 1709 | 12.32 | 2304 | 21.15 | 930 | 13.85 | 1212 |
| Qwen-2-7B | 46.05 | 739 | 7.37 | 434 | -6.48 | 1089 | 5.12 | 1413 | 16.33 | 679 | 13.68 | 871 |
| InternLM-2.5-7B | 45.16 | 666 | 3.17 | 430 | -9.84 | 1283 | 6.39 | 1431 | 19.64 | 911 | 12.91 | 944 |
| Mistral-7B-0.2 | 42.34 | 572 | 1.47 | 474 | -14.76 | 1222 | 13.05 | 869 | 7.88 | 606 | 10.00 | 749 |
| Phi-3.5-Moe | 54.27 | 629 | -3.70 | 609 | -10.01 | 1459 | -13.71 | 2444 | 13.95 | 737 | 8.16 | 1176 |
| MAP-Neo | 32.25 | 751 | 2.92 | 829 | -43.43 | 1086 | -9.02 | 924 | 17.45 | 824 | 0.03 | 883 |
| 🍋 Capability-Enhanced Large Language Models | | | | | | | | | | | | |
| LongWriter-GLM4-9B | 30.02 | **2679** | -35.01 | 439 | -5.57 | **4381** | 17.69 | **5257** | 34.53 | **3035** | 8.33 | **3158** |
| Suri-I-ORPO | 24.15 | 940 | -103.43 | **1233** | -118.06 | 2252 | -130.58 | 1770 | -89.91 | 1902 | -83.58 | 1619 |

same model family, API-based LLMs usually perform better than non-API-based LLMs (Yi-Large > Yi-1.5-34B), and LLMs with larger parameters show better performance (LLaMA-3.1-70B > LLaMA-3.1-8B). Among all LLMs, GPT-4o-2024-08-06 and Mistral-Large-API have the best long text generation results, with average scores exceeding 46, while Phi-3.5-Moe, MAP-Neo, and Suri-I-ORPO have the worst scores. Despite the better performance of GPT-4o-2024-08-06 and Mistral-Large-API, their scores remain around 50, indicating that there is still room for improvement.

(2) **Comparison of different tasks.** We also evaluate the long text generation capabilities of LLMs across various HelloBench tasks. Most LLMs perform poorly on summarization and chat tasks but achieve better results on open-end QA and text completion tasks. For instance, Claude-3.5-Sonnet scores 62.73 and 51.27 on open-end QA and text completion respectively, but only 31.34 on summarization and 32.60 on chat tasks. Additionally, the word count for summarization tasks is the lowest at around 500 words, while heuristic text generation and open-ended QA tasks have about 800 words, and chat and text completion tasks have the highest word count.

(3) **Analysis of word count.** Currently, most LLMs prefer to generate around 1,000 words for long text generation tasks when there are only implicit requirements in the instructions, such as {The article should be long enough to thoroughly explore the topic}. However, 1,000 words are often insufficient for many long text generation tasks. This means that when faced with long text generation tasks, current LLMs have a significant limit on word count or prefer to generate shorter text. Additionally, while capability-enhanced LLMs can generate significantly longer text, the overall quality of their generation decreases, resulting in lower scores.

(4) **Comparison of capability-enhanced LLMs.** By observing the results of LongWriter-GLM4-9B and Suri-I-ORPO, it is evident that these LLMs, enhanced for long text generation, can generate significantly longer text. However, the quality of the generated text has decreased, leading to low overall scores, especially for Suri-I-ORPO. Therefore, extending the output of LLMs while maintaining quality may be crucial to further improving long text generation capabilities.

Table 3: Length-Constrained Experiments, "w/o" represents without, "2K", "4K", "8K", and "16K" represent the requirements for LLMs to generate text over 2,000 words, 4,000 words, 8,000 words, and 16,000 words, respectively.

| Models | w/o constraint | | 2K | | 4K | | 8K | | 16K | |
|---|---|---|---|---|---|---|---|---|---|---|
| | S | WC | S | WC | S | WC | S | WC | S | WC |
| GPT-4o-2024-08-06 | **47.87** | 1121 | 7.05 | 1636 | -18.32 | 1949 | -78.03 | 1613 | -136.51 | 1368 |
| Claude-3.5-Sonnet | 40.92 | 941 | **39.55** | 2380 | **33.04** | 3846 | **18.74** | 5471 | -25.05 | 5549 |
| Mistral-Large-API | 47.07 | 859 | 24.36 | 1834 | -3.12 | 2329 | -57.19 | 2279 | -121.64 | 1390 |
| Yi-Large | 31.23 | 766 | -76.00 | 994 | -173.19 | 904 | -195.45 | 791 | -201.47 | 788 |
| LLaMA-3.1-70B | 31.84 | 910 | -19.97 | 1371 | -52.27 | 1531 | -82.28 | 1524 | -95.34 | 1661 |
| Qwen-2-72B | 35.66 | 740 | -42.34 | 1053 | -140.33 | 930 | -147.72 | 875 | -146.86 | 916 |
| InternLM-2.5-20B | 37.97 | 817 | -72.93 | 1050 | -95.98 | 1117 | -138.42 | 970 | -146.53 | 802 |
| LongWriter-GLM4-9B | 34.53 | **3035** | 6.68 | **3351** | 8.79 | **5279** | -3.11 | **8037** | -9.78 | **10010** |
| Suri-I-ORPO | -89.91 | 1902 | -165.22 | 2861 | -196.47 | 3035 | -209.11 | 3152 | -216.41 | 4405 |

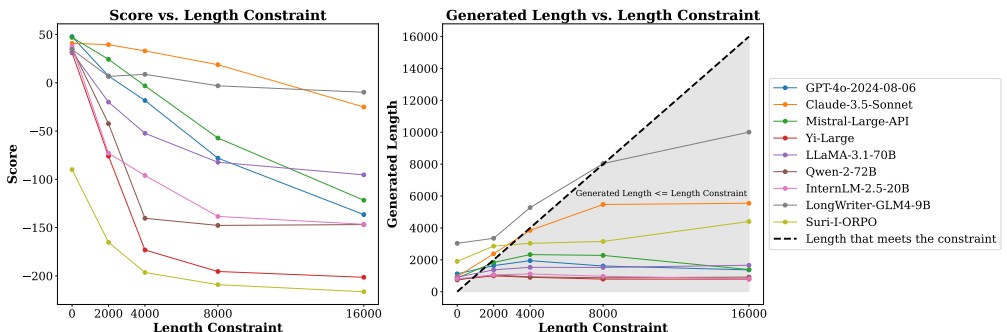

Figure 4: The scores and generated length of different LLMs under various length constraints. We consider "without constraint" as "length constraint = 0". The gray area on the right figure indicates regions where the generated lengths do not meet the length constraints.

### 4.3 LENGTH-CONSTRAINED EXPERIMENTS

In Section 4.2, we observe that LLMs prefer generating text around 1,000 words when there are no specific word count constraints. To further explore the generation quality and the limits of LLMs on output length, we have conducted length-constrained experiments. Specifically, we choose heuristic text generation task, with length constraints ranging from 2K to 16K, requiring LLMs to generate text exceeding these word counts. We add a length requirement to the original prompts and select a subset of LLMs, considering that many LLMs have a max_new_tokens[4] less than 16K. Within the same model family, we chose only one model as a representative.

The experimental results, shown in Table 3 and Figure 4, indicate a significant decrease in overall score as the length constraint increases. Among all LLMs, only Claude-3.5-Sonnet, LongWriter-GLM4-9B, and Suri-I-ORPO can generate text with more than 3,000 words, while other LLMs are limited to around 2,000 words. Claude-3.5-Sonnet performs the best, with a max_new_tokens limit of 8,192 tokens; while LongWriter-GLM4-9B also shows good performance, with the longest output among current LLMs. Despite the max_new_tokens of most LLMs reaching 16,384 tokens, it is still difficult for current LLMs to generate long text with explicit length constraints.

### 4.4 EFFECTIVENESS OF LONG-CONTEXT LLMS

Recently, many long-context enhancement methods have been used to extend the context window of LLMs, further improving their capabilities to understand long text. However, whether long-context LLMs perform well in generating high-quality long text remains an open question. To further explore it, we compare three mainstream open-source LLMs and their respective long-context-enhanced variants (Yi-1.5-34B-16K, InternLM-2.5-7B-1M, and GLM4-9B-Chat-1M). The experi-

---

[4] max_new_tokens is a generation parameter used to control the number of tokens generated by LLM, ensuring it does not exceed a certain value.

Table 4: Long-Context LLMs Ablation Study.

| Models | OEQA | | Summ | | Chat | | TC | | HTG | | AVG | |
|---|---|---|---|---|---|---|---|---|---|---|---|---|
| | S | WC | S | WC | S | WC | S | WC | S | WC | S | WC |
| Yi-1.5-34B | **47.36** | **751** | -14.33 | 328 | **5.02** | 1205 | **44.73** | 1054 | **35.31** | **875** | **23.63** | 843 |
| Yi-1.5-34B-16K | 46.25 | 678 | **11.78** | **449** | -6.56 | 1141 | -17.94 | **1706** | 28.48 | 795 | 12.40 | **954** |
| InternLM-2.5-7B | 45.16 | 666 | **3.17** | **430** | **-9.84** | 1283 | **6.39** | **1431** | 19.64 | **911** | **12.91** | **944** |
| InternLM-2.5-7B-1M | **49.15** | **708** | -17.43 | 330 | -25.83 | 1277 | 4.88 | 1160 | **23.01** | 803 | 6.76 | 855 |
| GLM-4-9B | **40.71** | **788** | -5.38 | 329 | **0.47** | 1709 | **12.32** | 2304 | **21.15** | 930 | **13.85** | 1212 |
| GLM-4-9B-1M | 38.07 | 724 | **1.21** | **342** | -54.92 | **2285** | -64.70 | **4049** | -25.55 | **3317** | -21.18 | **2144** |

mental results are shown in Table 4. In general, the quality of long text generated by LLMs with long-context enhancements is lower than that of the base LLMs, which indicates a negative correlation between LLMs' long-context understanding and their long text generation capabilities. For example, compared to Yi-1.5-34B, the score of Yi-1.5-34B-16K drop by an average of 11.23 points.

### 4.5 EFFECTIVENESS OF HELLOEVAL

Table 5: Spearman correlation coefficient and the corresponding p-value between different evaluation methods and human evaluation. The Spearman correlation coefficient is multiplied by 100.

| | HelloEval | LE | LE-C | AVG-C | METEOR | BLEU | ROUGE-L | R-4 | D-4 | PPL |
|---|---|---|---|---|---|---|---|---|---|---|
| **Spearman's** $\rho$ | **31.93** | 8.05 | 15.38 | 25.72 | 1.64 | -6.76 | -5.61 | -4.76 | 3.80 | 10.83 |
| **p-value** | 4.67e-7 | 3.33e-2 | 4.38e-5 | 7.99e-5 | 6.64e-1 | 7.37e-2 | 1.38e-1 | 2.08e-1 | 3.15e-1 | 4.12e-3 |

To demonstrate the effectiveness of HelloEval, we have conducted experiments to compare the evaluation results of different LLM-as-a-Judge evaluation methods and traditional metrics. (1) **Human Evaluation**: Based on the evaluation guideline in Appendix F, the human evaluation of the instructions and responses in HelloBench, serves as the ground truth for correlation computing. (2) **LLM-Eval (Zheng et al., 2024)**: Using GPT-4o to directly evaluate the response on a scale of 0-10, the prompt template is shown in Figure 10. (3) **LLM-Eval with Checklists (Lin et al., 2024)**: Based on LLM-Eval, we provide checklists and evaluate responses directly on a scale of 0 to 10, where the prompt template is shown in Figure 11. (4) **Average evaluation results of Checklists (Lee et al., 2024)**: Calculate the average of the evaluation results of the checklists given by LLM-as-a-Judge.

Details of other evaluation metrics are provided in Appendix J.2. Table 23 shows the evaluation results of different LLMs given by various evaluation methods or metrics, and Table 5 presents the Spearman correlation coefficient (Spearman, 1987) between different evaluation methods and human evaluation. A higher Spearman correlation coefficient indicates a stronger positive correlation, while a lower p-value signifies a more significant relationship. We find that HelloEval shows the highest correlation with human evaluation, indicating its effectiveness and alignment with humans. Additionally, traditional metrics are not suitable for evaluating long text generation, as their correlation with human evaluation is quite low, with some even showing a negative correlation.

## 5 ANALYSIS AND DISCUSSION

### 5.1 CURRENT CONCLUSIONS

The core conclusions of long text generation capabilities of LLMs are as follows:

**1. Short but Acceptable Quality**: Currently, most LLMs, when not constrained by specific word count, prefer to generate text that is around 1,000 words. The quality of the generated text at this length is acceptable, with GPT-4o and Mistral-Large-API performing the best. However, the scores still remain around the passing scores, indicating there is still room for improvement.

**2. Long but Low Quality**: Some LLMs that have enhanced long text generation capabilities (Suri-I-ORPO and LongWriter-GLM4-9B) can generate longer text around 3,000 words but the quality of the generated text decreases significantly.

**3. Limit in Word Count**: Although current LLMs have a max_new_tokens of 16,384 or more, they still struggle to generate such long text. In most cases, they prefer to generate text around 2,000 words when there are word count constraints. However, after training (SFT, DPO (Rafailov et al., 2024)), the length of the generated text can notably increase.

**4. Inherent Connections in Context Window**: Long-context LLMs' improved ability to understand long input doesn't necessarily enhance their long text generation capabilities. Nevertheless, there is an inherent connection, as both require an extended context window. Long-context LLMs can produce longer text in some tasks compared to standard versions, but often with a lower quality.

## 5.2 ERROR MODE ANALYSIS

After analyzing the error cases of different LLMs on HelloBench, we identify four main error modes, as shown in Figure 13: (1) **Repetition** - repetition when generating long. (2) **Rejection** - rejection for long text generation requests. (3) **Perception Error in Length** - LLMs incorrectly evaluate the word count of their generated text. (4) **Meaningless** - more meaningless text when generating longer text. We list here to help optimize LLMs further. The details are provided in Appendix K.

## 5.3 FUTURE RESEARCH DIRECTIONS AND DISCUSSIONS

We believe that future research could focus on enhancing the output length of LLMs while maintaining quality, addressing the potential trade-off between the two. Additionally, it is crucial to explore efficient methods beyond alignment training to shift from a long-input-short-output to a short-input-long-output paradigm. Furthermore, concurrently improving the understanding of long input and the generation of long output is essential for fully realizing LLMs' capabilities in handling long texts. The detailed discussions are provided in Appendix L.

## 6 RELATED WORKS

**Long-Context Capabilities of LLMs** Recently, many researchers have been focused on benchmarking the long-context capabilities of LLMs and exploring methods to enhance these capabilities. LongBench (Bai et al., 2023b) introduces the first bilingual, multi-task benchmark for long-context understanding, enabling a more rigorous evaluation of long-context understanding. LongIns (Gavin et al., 2024) proposes a challenging long-context instruction-based exam for LLMs, which is built based on the existing instruction datasets. In addition, there are many methods for enhancing long text capabilities based on RoPE (Peng et al., 2023; Chen et al., 2023).

**Long Text Generation Capabilities of LLMs** Long text generation capabilities are essential for LLMs, correlating with various real-world uses of LLMs, such as story generation (Venkatraman et al., 2024; Bai et al., 2024; Zhou et al., 2023), repository-level code completion (Liu et al., 2024b; Wang et al., 2024a), document generation (Luo et al., 2024), etc. To explore the long text generation capabilities of LLMs, ProxyQA (Tan et al., 2024) proposes an innovative framework to assess long text generation, LongWriter (Bai et al., 2024) develops LongBench-Write, a comprehensive benchmark for evaluating ultra-long generation capabilities. However, most of these benchmarks are not comprehensive, focusing only on a small part of long text generation scenarios.

## 7 CONCLUSION

In this paper, we introduce HelloBench, the first comprehensive, in-the-wild, and open-ended benchmark to evaluate long text generation capabilities of LLMs. First, we systematically categorize long text generation tasks using Bloom's Taxonomy, resulting in 5 tasks, 38 subcategories, and a total of 647 testing samples. Second, to evaluate the quality of long text generated by LLMs, we propose HelloEval, a human-aligned evaluation method for long text generation, which shows the highest correlation with human evaluation. Third, we observe that current LLMs still struggle to generate long text with high quality, and the generation length is also limited (around 2,000 words). We hope HelloBench could guide the developers and researchers to understand the long text generation capabilities of LLMs and facilitate the growth of foundation models.

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

## A EXAMPLES FOR EACH TASK IN HELLOBENCH

---

**An example of open-ended QA.**

You should write a detailed response to the following question on Science.

[Question]:
**Is the age old concept of turning a metal which is not Gold into Gold, known as Alchemy still fictional according to modern science? Doesn't nuclear science enable the creation of Gold isotopes from heavier elements?**

[Requirements]:
The answer should be long enough to provide a comprehensive response.

---

**An example of summarization.**

You're a professional wordsmith. Summarize the following news in a concise summary, ensuring that all essential information is included.

[Text Start]:
**Acknowledging that survivors of sexual violence often behave differently than victims of other crimes, researchers at the University of Texas at Austin released an expansive report Monday that the UT System will use to train hundreds of officers who handle campus sexual assaults.**
**The Blueprint for Campus Police, drafted by UT Austin's Institute on Domestic Violence and Sexual Assault, will be incorporated into training for almost 600 officers across all eight of the system's academic institutions.**
**"Police in America, historically, have responded to the investigation of crimes in kind of a generalized fashion, regardless of whether it's a homicide, robbery, theft," or assault, according to Mike Heidingsfield, the UT System director of police. Because assault victims have experienced trauma, their cases often call for a more specialized officer response he said.**
**The training is especially necessary because of the prevalence of sexual assault, according to Noël Busch-Armendariz, the report's principal investigator. One study, released in September, found that more than 18 percent of female undergraduates at UT Austin had been sexually assaulted since arriving on campus.**
**The report offers specific guidelines for officers from the moment they first interact with victims. "Let the victim know that they are safe," the report reads. "Let the victim know they will not be judged," and "understand that a victim's alcohol or drug use is an issue of increased vulnerability rather than culpability."**
...
[Text End]

[Requirements]:
1. Identify the main theme and core assertions of the article.
2. Extract key supporting details, statistics, and data.
3. Ensure the summary accurately includes all essential points and correct information, without adding any details not present in the original text.
4. Capture important quotes from key individuals.
5. Maintain the original meaning and tone without personal opinions.
6. Preserve the chronological order of events if applicable.
7. Provide a long summary to contain all the needed information.

---

An example of **chat**.

**Write a business plan for a new non profit org. The non profit org will address the digital divide in urban communities. Write in great detail about Executive summary, Nonprofit description, Need analysis, Products, programs, and services descriptions. The non profit will offer free tech training to qualifying individuals. Outline the goals and objectives to achieve our mission, Operational plan, Marketing plan, Impact plan, and Financial plan. How to build awareness for the cause. How to raise funds from donors. Funding sources: List out grants and significant funds you've received.Fundraising plan: Outline how you plan to raise additional funds. The organization plans to go from local, to. international once fully established. Be very detailed in all aspects. Each description should be very detailed.**

---

An example of **text completion**.

You should write a continuation of the following story.

[Story]:
**After the destruction of an energy world at the hands of Jacques Marcus, He decides to go to a hub-world on the other side of the system to recuperate and gear up for his next battle. Little does he know, the next battle is not far behind.**
**Jacques arrives on a planet that looks similar to Earth in every way except it's bigger. The city he lands in is the capital of the world named Solis City. He finds a map of the city at the port dock where his ship the Raging Phoenix is at. He makes his way to an Armory that's close to the dock. He enters the ramshackle building and talks to the wild looking shopkeeper. The shopkeeper says "Welcome to Pinpoint, the highest rated gun shop among tourists."**
**Jacques responds as he looks around the shop. "I highly doubt that."**
**"Well rude guy, anything you in the market for? My name's Keith by the by, what's yours stranger?"**
**...**

[Requirements]:
1. The continuation should be consistent with the original story in terms of plot, character development, and tone.
2. Maintain coherence and logical progression in the storyline.
3. Ensure the continuation is long enough to cover the necessary developments.

---

An example of **heuristic text generation**.

You should write an engaging story based on the following writing prompt.

[Writing Prompt]:
**You got one wish, and it was for immortality. It only took a few years to realize you no longer age, but you only just found out you're not unkillable, but circumstances will change around you to prevent you from getting hurt.**

[Requirements]:
1. Feel free to use creativity to expand on the prompt and create an interesting and captivating narrative.
2. Ensure the story is long enough.

# B   DETAILS OF DATASET COLLECTION

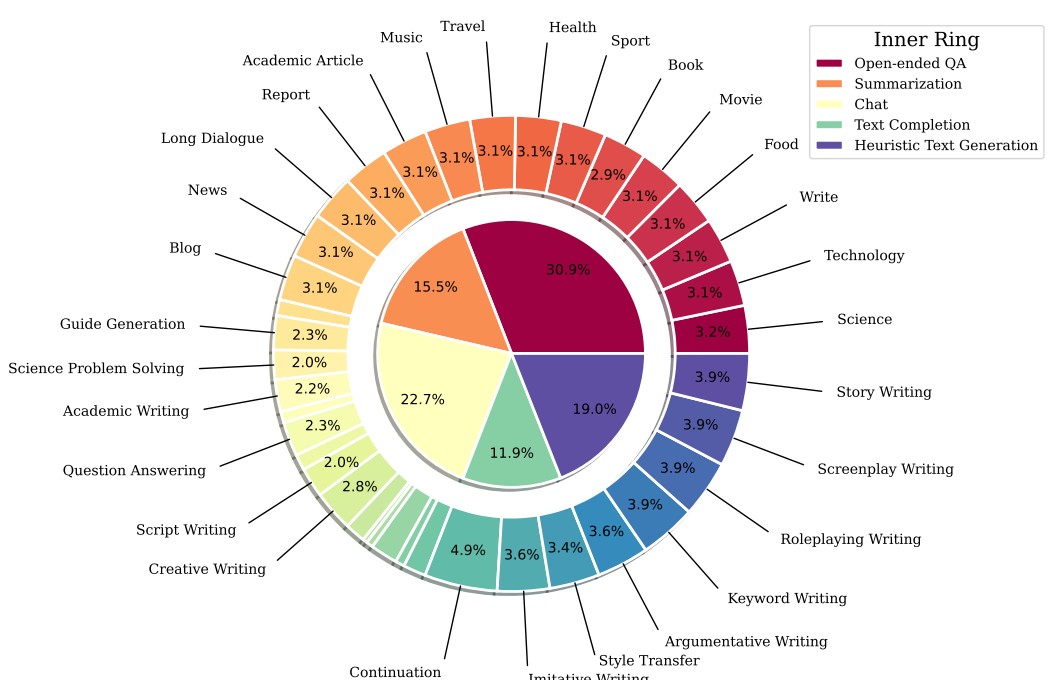

Figure 5: HelloBench Categories and Subcategories Distribution: The inner ring shows the categories and their proportions within the HelloBench. The outer ring details the subcategories and their respective proportions relative to the HelloBench.

**Open-Ended QA**   Quora is a question-and-answer website where people can ask questions and get answers from the community, users can share knowledge, opinions, and experiences on a wide range of topics. Quora is very suitable as a data source for open-ended QA. First, it consists of questions asked by users in real scenarios, with each question receiving different responses, making it an open-ended question that requires a longer response. Second, the questions are original, created by users, and can be filtered by time, making it ideal for collecting the latest and most original data. Additionally, Quora allows for topic-wise classification, providing natural subcategories by topic. Specifically, we selected the 10 most popular topics (Technology, Sport, Movie, Book, Music, Food, Health, Writing, Science, Travel) as our core data sources. Quora allows questions to be sorted by chronological order and provides metrics for question popularity (number of responses, likes, etc.). To ensure the novelty and timeliness of the questions, we first collected around 40 of the latest and most popular questions from each topic (with a cutoff date of July 19, 2024). We then filtered our data based on the following four criteria: (1). Retain questions that are suitable for long responses, we can refer to the length of user responses to the question on Quora. (2). Remove questions related to current events, such as reviews of the latest movies or news. (3). Exclude questions related to politics, gender, and sensitive content. (4). Remove semantically similar data.

**Summarization**   As a classic task in natural language processing, to ensure the practicality and diversity of summarization tasks in the field of long text generation, we have decided to select samples from publicly available datasets. By doing so, we guarantee that the probability of data leakage is minimized, and the quality of these data is ensured. To be specific, the summarization task in HelloBench is actually the long summarization task. We need LLMs to retain most key information while summarizing, and the compression rate of the original document should be larger than what is typically required in a general summarization task. We have gathered seven public datasets and divided them into these 5 subcategories:

- **News Summarization**: We collected data from Multi-News (Fabbri et al., 2019), which consists of news articles and human-written summaries sourced from newser.com[5].

- **Blog Summarization**: For the blog summarization task, the data includes sources from Reddit (Hamilton et al., 2017) and WikiHow (Koupaee & Wang, 2018). The Reddit dataset comprises various posts, while the WikiHow dataset is constructed from the online knowledge base available at wikihow.com[6].

- **Long Dialogue Summarization**: We collected data from QMSum (Zhong et al., 2021), which contains multi-domain meeting records.

- **Report Summarization**: Our dataset for this subcategory is sourced from GovReport (Huang et al., 2021), consisting of reports authored by government research agencies.

- **Academic Article Summarization**: We collected academic articles from PubMed (Sen et al., 2008) and Arxiv (Cohan et al., 2018), covering a wide range of topics including physics, medicine, and biology.

For each publicly available dataset, we have selected samples from the test and validation sets that have original text lengths between 3,000 and 6,000 words. The choice of this word range serves two purposes. First, it ensures the text is long enough so that the LLM naturally produces a longer summary. Second, it keeps the text from being too long, reducing evaluation pressure and ensuring HelloBench is suitable for more models, as many LLMs have a context window of 16k. Domain experts then review these samples to remove any texts that are obviously low-quality, such as those containing indecipherable formulas or those are obviously garbled text from OCR of PDF. Finally, we retain 20 samples for each subcategory.

**Chat**   We adhere to the steps below to collect and process data for the chat tasks in HelloBench:

- **Step 1:** We selected data from the WildChat, using NLTK (Loper & Bird, 2002) for word segmentation and filtering out conversations where the model's responses exceeded 1,000 words. We filtered out data flagged as toxic or redacted, keeping only the conversations labeled as "English".

- **Step 2:** To deduplicate the instructions, we used the BM25 algorithm (Robertson et al., 2009) to identify the top-5 most similar instructions for each entry. If two instructions are on each other's top-5, they are considered similar, and only one is kept. Additionally, we observed that instructions sharing the same first 25 characters are often similar, so we also removed these as duplicates.

- **Step 3:** To label the conversations, we first generated tags for each instruction using GPT-4o. These tags were normalized by converting them to lowercase and applying NLTK's WordNetLemmatizer to convert all tags back to their base forms. The normalized tags were then vectorized using PhraseBERT (Wang et al., 2021) and clustered using the DBSCAN algorithm (Khan et al., 2014). The purpose of clustering is to merge similar tags into parent categories. Otherwise, having too many tags will make the collection unmeaningful. Given the challenge of achieving optimal clustering in a single iteration, we performed iterative clustering. In each iteration, the tags from the same cluster identified in the previous iteration are concatenated with commas for vector encoding, and we perform a new iteration of DBSCAN clustering. Once a cluster reaches a threshold of 200 tags, it is considered as a final cluster and assigned a category name, while the remaining tags proceed through additional iterations. After labeling and clustering, GPT-4o was utilized to filter out instructions that were of low quality or those that did not match their assigned categories.

- **Step 4:** Domain experts carefully checked and selected instructions, while ensuring a sufficient number of categories are retained.

We end up with 147 instructions, which we categorize into 15 categories: *report write*, *guide generation*, *science problem solve*, *academic write*, *continue write*, *question answering*, *rewrite*, *script write*, *creative write*, *idea generation*, *explanation*, *data analysis*, *character creation*, *curriculum development*, and *question generation*. It's important to highlight that the subcategories for the chat

---

[5]https://www.newser.com/

[6]http://www.wikihow.com/

task in HelloBench come from the clustering results in Step 3. Additionally, we finally create 76 normalized categories and display the data proportions corresponding to each category in Figure 6. We then selected 15 representative categories as subcategories of chat tasks in HelloBench.

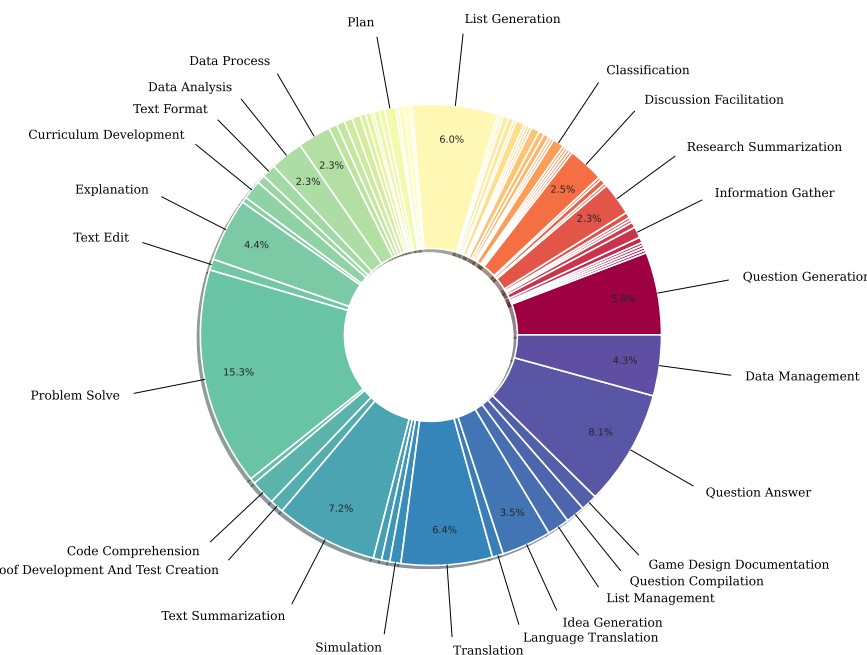

Figure 6: The distribution of categories after labeling in Step 1 and clustering in Step 3.

During data collection, we found that around 4% of WildChat's English conversations, which are neither toxic nor redacted, included responses longer than 1,000 words. Though this percentage is not significant, the actual percentage might be underestimated. WildChat shared a website while collecting data, and we think that most users of this site are likely researchers or enthusiasts in the NLP field. As a result, they may already know that LLMs struggle to generate longer text and thus have reduced the proportion of instructions for long text generation. Additionally, many instructions may request LLMs to produce longer text, but LLMs like GPT-4o, which are aligned through RLHF (Ouyang et al., 2022), are prone to reject such instructions, leading to some instructions being ignored. Therefore, we believe that the demand for long text generation is actually higher than 4%. Nevertheless, filtering for responses exceeding 1,000 words almost guarantees that the instructions collected are suitable for long text generation scenarios. Thus, our data collection approach is reasonable.

**Text Completion** The reasons for choosing continuation, imitation, and style transfer as text completion subcategories are that these three tasks are very natural text completion tasks, and we observed from WildChat (Zhao et al., 2024) that they have real-world scenarios. We collected around 200 stories from the subreddit r/shortstories. Among these stories, some are unfinished, making them suitable for continuation tasks. Additionally, each story has a corresponding topic, thus the imitation task is defined as writing a story on the topic in the preceding story's style. For the style transfer task, we pre-defined 10 different writing styles. The style transfer task is defined as converting the current story into a new style. Besides, the ten pre-defined writing styles for the style transfer task are:

1. **Hemingwayesque**: Characterized by concise, straightforward prose, minimalistic descriptions, and an emphasis on dialogue.

2. **Dickensian**: Features detailed descriptions, complex characters, and social commentary, often with a focus on the struggles of the poor.

3. **Joycean**: Known for stream-of-consciousness technique, intricate wordplay, and deep exploration of characters' inner thoughts.

4. **Austenian**: Combines witty, satirical commentary on society with a focus on romantic relationships and character development.

5. **Faulknerian**: Utilizes long, complex sentences, multiple perspectives, and a deep sense of place, often set in the American South.

6. **Proustian**: Rich, detailed prose that delves into memory and perception, often with long, flowing sentences.

7. **Woolfian**: Emphasizes stream-of-consciousness narrative, lyrical prose, and deep psychological exploration of characters.

8. **Lovecraftian**: Features cosmic horror, elaborate mythologies, and a sense of existential dread, often with archaic language.

9. **Kingian**: Combines everyday settings and relatable characters with elements of horror, suspense, and supernatural phenomena.

10. **Kafkaesque**: Features surreal, nightmarish scenarios, often with themes of alienation and absurdity.

Additionally, the four criteria for filtering stories are as follows:

1. The length of the stories should be between 1,000 and 5,000 words, requiring LLMs to generate longer completions implicitly.

2. The stories should be as diverse as possible in terms of topic.

3. Remove stories containing sensitive information.

4. Remove semantically similar stories.

**Heuristic Text Generation**    The five pre-defined subcategories and their collection methods are as follows:

1. **Story Writing**: Given a story writing prompt, LLMs are asked to create a complete story. Writing prompts are sourced from r/WritingPrompts[7] on Reddit where users share creative writing prompts to inspire stories and other written works. We collected the latest 40 writing prompts (dates: July 10, 2024 - July 12, 2024) along with user responses. We deduplicated the writing prompts and then manually filtered the writing prompts for quality, retaining those with longer responses and ensuring diversity among the writing prompts.

2. **Keyword Writing**: Given a topic and corresponding keywords, LLMs are asked to write an article about the topic and keywords. The keywords were generated by GPT-4o, producing 30 different topics and keywords. We then filtered and retained around 25 high-quality topics and keywords.

3. **Argumentative Writing**: Given an argumentative topic, LLMs are asked to write an argumentative essay on it. The topics are sourced from the New York Times[8]. The "Learning Student Opinion" section of The New York Times is a platform where students can express their views on various topics. It features prompts related to current events, social issues, and other subjects, encouraging students to engage critically and thoughtfully. This section aims to foster discussion and reflection among young people, providing a space for them to share their perspectives and develop their voices. We collected all the topics from March 2024 to May 2024 and filtered the data based on the following criteria: (1). Remove topics strongly related to current events. (2). Remove topics directly related to students' personal experiences. (3). Based on student responses, retain topics suitable for long text generation. (4). Remove semantically similar topics. After filtering, we kept 23 topics.

4. **Screenplay Writing**: Given screenplay writing prompts, LLMs are asked to write a complete screenplay. The main difference from story writing is that screenplay writing is more

---

[7]https://www.reddit.com/r/WritingPrompts/
[8]https://www.nytimes.com/column/learning-student-opinion

structured and requires consideration of character information for each scene. The screenplay writing prompts are sourced from Squibler[9], which lists 61 interesting prompts. Similarly, we deduplicated prompts and then filtered these prompts to retain high-quality and those suitable for long text generation while ensuring the prompts are diverse.

5. **Roleplaying Writing**: Given writing prompts, write a complete story from the character's first-person perspective. The prompts are sourced from a blog[10] that lists 77 useful prompts for roleplaying writing. Similarly, we deduplicated the prompts and then manually filtered these prompts to retain diverse, high-quality, and suitable data for long text generation.

**Prompt Wrapping**  After collecting data for the five tasks in HelloBench, we need to perform prompt wrapping before evaluating LLMs. This step is essential because it affects how well LLMs understand the instructions and finally influences the evaluation results. To be specific, our prompt wrapping is shown in Figure 7.

---

Prompt Wrapping for HelloBench

[Simple Task Definition (usually in one sentence)]

[Collected Data (e.g., source documents in summarization tasks, stories in text completion tasks, etc.)]

[Necessary Requirements]

---

Figure 7: Prompt Wrapping for HelloBench

Prompt Wrapping for HelloBench consists of three main parts. The first part is a simple definition of the task, usually one sentence long. The second part includes the collected data, which is typically necessary for the task, such as source documents for summarization tasks. The third part is some necessary requirements for the instructions. For example, in summarization tasks, we require that LLMs generate sufficiently long summarization to cover the main points of the source documents, suitable for long text generation tasks. Prompt Wrapping for HelloBench primarily targets the chat versions of LLMs. We do not need to wrap prompts for chat tasks because they are naturally chat prompts. For specific examples, please refer to Appendix A.

## C  DATA QUALITY OF HELLOBENCH

To demonstrate the data quality of the HelloBench, we have conducted additional experiments. Specifically, we validated two aspects of quality. First, we evaluate whether the data in HelloBench is inherently suitable for long text generation, rather than merely adding a requirement for longer output on the task. Second, we simply explain the data leakage problem within the HelloBench.

For the first part, we hired 3 annotators to make a simple binary judgment on the instructions in HelloBench, specifically whether they believe the response to a given instruction should exceed 1,000 words. To prevent evaluators from having prior biases, we did not reveal the purpose of our evaluation beforehand. Tables 6 and 7 show the rates of responses they believe exceed 1,000 words and the correlation scores among three annotators. "HM1", "HM2", and "HM3" represent three annotators.

Data leakage is a crucial problem to consider in the evaluation benchmarks for LLMs (Wang et al., 2024d). Since LLMs are trained on vast amounts of web data during the pretraining stage, there is a significant risk that the test data may already be included in the pretraining data. Benbench (Xu et al., 2024) highlights this problem by comparing the ppl of different LLMs on the training and test sets of the GSM8K and MATH datasets.

---

[9]https://www.squibler.io/learn/writing/writing-prompts/dialogue-prompts/
[10]https://robinpiree.com/blog/roleplay-prompts

Table 6: The rate given by three annotators.

| Annotator | Rate |
|-----------|-------|
| HM1 | 89.49 |
| HM2 | 87.64 |
| HM3 | 85.63 |

Table 7: Pearson Correlation Coefficient Among three Annotators.

|  | HM1 | HM2 | HM3 |
|-----|--------|--------|--------|
| HM1 | 1 | 0.7439 | 0.8364 |
| HM2 | 0.7439 | 1 | 0.7696 |
| HM3 | 0.8364 | 0.7696 | 1 |

For HelloBench, however, we adhered to the principles of open-ended and timeliness when collecting data. We collected real user data to ensure originality and made efforts to collect the latest data available online, minimizing the possibility of data leakage. Moreover, HelloBench focuses on evaluating text generation, which involves open-ended text evaluation rather than having a correct answer or ground truth. Therefore, even if some data leakage might occur, we believe it would not significantly impact the evaluation results. HelloEval evaluates the quality, factuality, and completeness of the generated text, which differs from standard evaluation. In summary, we believe that HelloBench will not face serious data leakage issues, even as time progresses.

# D  ADDITIONAL MATERIALS FOR DATASET STATISTICS

Table 8: The number of each category and corresponding subcategories in HelloBench.

| Category | Nums |
|---|---|
| Open-Ended QA | 200 |
| # science | 21 |
| # technology | 20 |
| # write | 20 |
| # food | 20 |
| # movie | 20 |
| # book | 19 |
| # sport | 20 |
| # health | 20 |
| # travel | 20 |
| # music | 20 |
| Summarization | 100 |
| # academic article | 20 |
| # report | 20 |
| # long dialogue | 20 |
| # news | 20 |
| # blog | 20 |
| Text Completion | 77 |
| # continuation | 32 |
| # imitative writing | 23 |
| # style transfer | 22 |

| Category | Nums |
|---|---|
| Chat | 147 |
| # report writing | 7 |
| # guide generation | 15 |
| # science problem solving | 13 |
| # academic writing | 14 |
| # continue writing | 5 |
| # question answering | 15 |
| # rewrite | 7 |
| # script writing | 13 |
| # creative writing | 18 |
| # idea generation | 9 |
| # explanation | 2 |
| # data analysis | 3 |
| # character creation | 12 |
| # curriculum development | 4 |
| # question generation | 10 |
| Heuristic Text Generation | 123 |
| # argumentative writing | 23 |
| # keyword writing | 25 |
| # roleplaying writing | 25 |
| # screenplay writing | 25 |
| # story writing | 25 |

Figure 8: Illustration of Word Lengths of Instructions in HelloBench

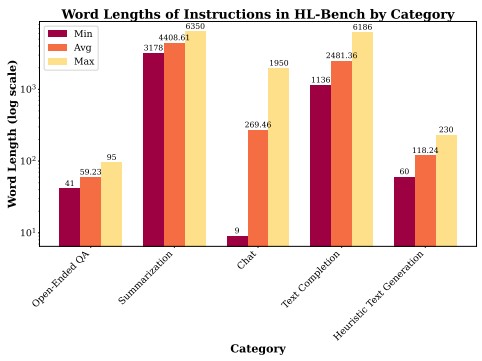

Table 9: Word Lengths of Instructions in HelloBench.

| Category | Min. | Max. | Avg. |
|---|---|---|---|
| Open-Ended QA | 41 | 95 | 59.23 |
| Summarization | 3178 | 6350 | 4408.61 |
| Chat | 9 | 1950 | 269.46 |
| Text Completion | 1136 | 6186 | 2481.36 |
| Heuristic Text Generation | 60 | 230 | 118.24 |

# E  CONSTRUCTION OF CHECKLISTS

As mentioned in Section 2.1, HelloBench is two-level classified. The first level includes categories such as open-ended QA and heuristic text generation. The second level further classifies open-ended QA into subcategories like Science and Technology. The checklists are designed specifically for each of these subcategories. Specifically, for a given subcategory like **open-ended QA – science**, (1) We first investigate the related works on its parent category (which is open-ended QA) and summarize the evaluation criteria from these works. (2) Then, we invite 5 domain experts to review the data of this subcategory in HelloBench and summarize 3-5 evaluation criteria they consider important. (3) We collect these evaluation criteria together and remove similar ones. (4) After that,

we ask 10 annotators to vote on the most important evaluation criteria, and then we only keep the top 4-6 evaluation criteria based on the votes of annotators. (5) Finally, we expand these criteria into yes-or-no checklists by using a powerful LLM (GPT-4o (OpenAI, 2024)).

## E.1 OPEN-ENDED QA

---

**The Checklists for Open-Ended QA**

---

1. Does the response content not only directly address the question but also ensure that every part of the response is strictly related to the topic of the question? Evaluate each sentence and paragraph rigorously to confirm it is entirely relevant to the topic and does not deviate in any way. If the question asks for personal feelings or opinions, the response must thoroughly provide the corresponding content. If even a single part of the response is slightly unrelated, redundant, or lacking in personal perspective when required, you must consider the response as not directly answering the question.

2. Is every aspect of the response impeccably factually correct? For instance, when listing historical information, are all mentioned historical figures, dates, and events precisely accurate? When presenting scientific terms or phenomena, are they completely factually accurate and up-to-date? Every word and paragraph of the response must undergo meticulous evaluation to ensure absolute factual correctness. If any single part of the response contains even a minor factual error or shows any uncertainty in its statements, you must consider the response as not factually correct.

3. Is the content of the response easy to understand? For difficult-to-understand technical terms, are there corresponding explanations and examples provided? Are more complex terms replaced with simpler ones? Every part of the response should be easy to understand, evaluated word by word and paragraph by paragraph. If there is any content you think can be optimized to be more concise or easier to understand, you should consider the response not easy to understand.

4. Is the content of the response interesting or novel? Because the questions are open-ended, the responses can be varied. An excellent response should present unique viewpoints or interesting content. Does the response offer a fresh perspective? If not, you should consider the response uninteresting.

5. Is the content of the response exceptionally rich and detailed, with no fewer than 500 words? Does each point include multiple, well-explained examples or explanations for strong support? If any part of the response is perceived as not thoroughly detailed or if any point lacks sufficient examples or explanations, you must consider the response incomplete and not lengthy.

6. Is the content of the response human-like? The content should not appear to be machine-generated. Evaluate each sentence and paragraph. Human responses usually do not have strange structures, such as markdown-like titles and subtitles. Human responses are generally flowing and may include many personal phrases like "I think" or other expressions of personal color. When making your judgment, you should forget the premise that the response is model-generated. Evaluate it without any prior bias. If you think it even slightly resembles machine-generated content, you should judge it as not human-generated.

7. Is the response flawless? If you think there is room for improvement, you should not consider the response flawless.

---

Table 10: The Checklists for Open-Ended QA.

Inspired by (Stelmakh et al., 2022; Malaviya et al., 2023; Hou et al., 2024), we have summarized the following evaluation criteria for open-ended QA:

1. **Coverage**: Evaluate whether the response comprehensively covers the key points of the question.

2. **Redundancy**: Evaluate whether the response includes irrelevant content that does not relate to the question.

3. **Consistency**: Evaluate if the response follows a natural and smooth logical flow.

4. **Accuracy**: Evaluate if the response uses reliable and accurate information rather than hallucination.

5. **Depth**: Evaluate whether the response includes sufficient and targeted details, rather than being overly general.

For open-ended QA, we realized that designing specialized checklists for each question domain (Technology, Sport, Movie, Book, Music, Food, Health, Writing, Science, Travel) is unnecessary.

The votes related to the domain-specific checklists are relatively few because the domain of a question does not significantly impact the final evaluation. Therefore, we set the same checklists for all subcategories of open-ended QA. When designing the checklists, we ensure each checklist is as complex as possible to provide better differentiation in the evaluation results of LLM-as-a-Judge. The final checklists for open-ended QA are listed in Table 10.

E.2 SUMMARIZATION

---

**The Checklists for Summarization (Part 1)**

Subcategory: **News Summarization.**
1. Is the content of the summary easy to understand? For difficult-to-understand technical terms, are there corresponding explanations and examples provided? Are more complex terms replaced with simpler ones? Every part of the summary should be easy to understand, evaluated word by word and paragraph by paragraph. If there is any content you think can be optimized to be more concise or easier to understand, you should consider the summary not easy to understand.
2. Is the summary sufficiently long and complete? Since the original news is lengthy, the summary should also be long enough to cover the key information from the news.
3. Is the summary perfectly accurate and unbiased? Every statement in the summary must strictly match the original news, with no additions, no deviations and no personal opinions. All statistical information and data must be identical to those in the original news. Even the slightest inconsistency or any additional information not present in the original news should make the summary be considered inaccurate.
4. Does the summary comprehensively cover all the important information from the original news, including when and where the news took place, who was involved, and what happened?
5. Does the summary perfectly meet all the requirements specified in the user instruction?
6. Do you think this summary is flawless? You should determine the checklist score based on whether there is room for improvement in the summary.

---

Subcategory: **Blog Summarization.**
1. Is the content of the summary easy to understand? For difficult-to-understand technical terms, are there corresponding explanations and examples provided? Are more complex terms replaced with simpler ones? Every part of the summary should be easy to understand, evaluated word by word and paragraph by paragraph. If there is any content you think can be optimized to be more concise or easier to understand, you should consider the summary not easy to understand.
2. Is the summary sufficiently long and complete? Since the original blog is lengthy, the summary should also be long enough to cover the key information from the blog.
3. Is the summary perfectly accurate without inserting personal opinions? Every statement in the summary must strictly match the original blog, with no additions or deviations. All statistical information and data must be identical to those in the original blog. Even the slightest inconsistency or any additional information not present in the original blog should make the summary be considered inaccurate.
4. Does the summary comprehensively cover all the important information from the original blog, including main topic, primary arguments, details that support the arguments.
5. Does the summary perfectly meet all the requirements specified in the user instruction?
6. Do you think this summary is flawless? You should determine the checklist score based on whether there is room for improvement in the summary.

---

Table 11: The Checklists for Summarization (Part 1).

Inspired by (El-Kassas et al., 2021), we have summarized the following evaluation criteria for summarization:

1. **Coverage**: The summarization includes the key information present in the source document.

2. **Redundancy**: The summarization avoids unnecessary repetition, such as repeated sentences or overused noun phrases.

3. **Readability**: The summarization is fluent and easily understandable, with clear logic and well-organized information.

---

**The Checklists for Summarization (Part 2)**

Subcategory: **Academic Article Summarization.**
1. Is the content of the summary easy to understand for a general academic audience? For difficult-to-understand technical terms, are there corresponding explanations and examples provided? Are more complex terms replaced with simpler ones? Every part of the summary should be easy to understand, evaluated word by word and paragraph by paragraph. If there is any content you think can be optimized to be more concise or easier to understand, you should consider the summary not easy to understand.
2. Is the summary sufficiently long and complete? Since the original article is lengthy, the summary should also be long enough to cover the key information from the article.
3. Is the summary perfectly accurate without errors or misleading information? Every statement in the summary must strictly match the original article, with no additions or deviations. All statistical information and data must be identical to those in the original article. Even the slightest inconsistency or any additional information not present in the original article should make the summary be considered inaccurate.
4. Does the summary comprehensively cover all the important information from the original article, including research background, methods, findings, reuslts and conclusions?
5. Does the summary perfectly meet all the requirements specified in the user instruction?
6. Do you think this summary is flawless? You should determine the checklist score based on whether there is room for improvement in the summary.

---

Subcategory: **Report Summarization.**
1. Is the content of the summary easy to understand? For difficult-to-understand technical terms, are there corresponding explanations and examples provided? Are more complex terms replaced with simpler ones? Every part of the summary should be easy to understand, evaluated word by word and paragraph by paragraph. If there is any content you think can be optimized to be more concise or easier to understand, you should consider the summary not easy to understand.
2. Is the summary sufficiently long and complete? Since the original report is lengthy, the summary should also be long enough to cover the key information from the report.
3. Is the summary perfectly accurate? Every statement in the summary must strictly match the original report, with no additions or deviations. All statistical information and data must be identical to those in the original report. Even the slightest inconsistency or any additional information not present in the original report should make the summary be considered inaccurate.
4. Does the summary comprehensively cover all the important information from the original report, including key statistical information, recommendations, and conclusions?
5. Does the summary perfectly meet all the requirements specified in the user instruction?
6. Do you think this summary is flawless? You should determine the checklist score based on whether there is room for improvement in the summary.

---

Subcategory: **Long Dialogue Summarization.**
1. Is the content of the summary easy to understand? For difficult-to-understand technical terms, are there corresponding explanations and examples provided? Are more complex terms replaced with simpler ones? Every part of the summary should be easy to understand, evaluated word by word and paragraph by paragraph. If there is any content you think can be optimized to be more concise or easier to understand, you should consider the summary not easy to understand.
2. Is the summary sufficiently long and complete? Since the original dialogue is lengthy, the summary should also be long enough to cover the key information from the dialogue.
3. Is the summary perfectly accurate without error or misleading information? Every statement in the summary must strictly match the original dialogue, with no additions or deviations. All statistical information and data must be identical to those in the original dialogue. Even the slightest inconsistency or any additional information not present in the original dialogue should make the summary be considered inaccurate.
4. Does the summary comprehensively cover all the important information from the original dialogue, including key topics discussed and every role's viewpoint?
5. Does the summary thoroughly exclude all redundant information, filler words, unnecessary rhetoric, and irrelevant interjections without omitting any key points or altering the original meaning and context of the conversation?
6. Does the summary perfectly meet all the requirements specified in the user instruction?
7. Do you think this summary is flawless? You should determine the checklist score based on whether there is room for improvement in the summary.

Table 12: The Checklists for Summarization (Part 2).

4. **Accuracy**: The summarization accurately reflects the source document without errors or misleading information, with each piece of summarization coming from the source document.

Table 11 and Table 12 present the checklists for summarization tasks.

### E.3 CHAT

---

**The Checklists for Chat**

---

**General Checklists** for all Chat data:

1. Does the response fully comprehend all specific aspects of the user's instructions and accurately address each requirement with thoroughness and precision, ensuring it strictly meets the user's needs without any omissions or misunderstandings?

2. Is the response sufficiently long and comprehensive, addressing all aspects of the user's instructions with detailed and complete information, ensuring no part of the requirement is overlooked?

3. Is the content of the response easy to understand? For difficult-to-understand technical terms, are there corresponding explanations and examples provided? Are more complex terms replaced with simpler ones? Every part of the response should be easy to understand, evaluated word by word and paragraph by paragraph. If there is any content you think can be optimized to be more concise or easier to understand, you should consider the response not easy to understand.

4. Is every aspect of the response impeccably factually correct? For instance, when listing historical information, are all mentioned historical figures, dates, and events precisely accurate? When presenting scientific terms or phenomena, are they completely factually accurate and up-to-date? Every word and paragraph of the response must undergo meticulous evaluation to ensure absolute factual correctness. If any single part of the response contains even a minor factual error or shows any uncertainty in its statements, you must consider the response as not factually correct.

5. Do you think this response is flawless? You should determine the checklist score based on whether there is room for improvement in the response.

---

**Specific Checklist** for each subcategory of Chat:

**Script Writing**: Does the generated script contain detailed script-specific structured information, including scene settings, transitions between acts, character actions, and expressions, ensuring that each element is clearly defined and contributes to the overall coherence and flow of the script?

**Idea Generation**: Is the generated idea highly creative and truly original, presenting a concept that is neither obvious nor easily conceived by others? Additionally, does the idea stand out as unique and unprecedented, ensuring it has not been previously thought of or widely recognized?

**Curriculum Development**: Does the curriculum comprehensively cover all key learning objectives, ensuring each objective is addressed with depth and clarity, and is supported by well-structured lessons, activities, and assessments that reinforce understanding and application?

**Character Creation**: Are the created characters exceptionally interesting, possessing unique and multi-dimensional traits, richly developed backgrounds, consistently captivating actions and motivations, and a significant and integral contribution to the storyline that enhances the overall narrative depth and engagement?

**Report Writing**: Does the report avoid appearing machine-generated, looking like it was written by a human, and refraining from using overly structured language and overly concise content? If you think it even slightly resembles machine-generated content, you should judge it as not human-generated.

**Guide Generation**: Is the generated guide highly useful, providing clear, detailed, and easy-to-follow step-by-step instructions that effectively address all potential user questions and issues?

**Academic Writing**: Does the response comprehensively cover all the important and detailed information, including research background, methods, findings, results and conclusions?

**Rewrite**: Does the rewritten content remain fully consistent with the original content, accurately preserving all key points, nuances, and context, while enhancing clarity and readability without any loss of meaning, important information, or original intent?

**Data Analysis**: Are the data findings not only accurately interpreted but also thoroughly analyzed, with all interpretations clearly supported by the data and contextualized within the broader research or study framework?

**Explanation**: Is the explanation exceptionally easy to understand, with each part thoroughly and clearly explained, ensuring no ambiguity or confusion for the user?

**Creative Writing**: Is the generated content highly novel and creative? An excellent response should present unique viewpoints or interesting content. Does the response offer a fresh perspective? If not, you should consider the response not creative.

**Question Answering**: Does the response address all questions mentioned in the instructions, providing relatively complete answers to each one?

**Continue Writing**: Is the continuation not only consistent with the preceding text but also seamlessly integrated, maintaining logical flow, coherence, and alignment with the established tone and context?

**Science Problem Solving**: Are all the reasoning steps, mathematical formulas, and calculations mentioned in the response not only completely correct but also clearly explained and easy to understand, ensuring no ambiguity or confusion for the user?

**Question Generation**: Does the number of generated questions meet the requirements, with each question being unique and representative, and is there no repetition among the different questions?

---

Table 13: The Checklists for Chat.

The chat task of HelloBench is sourced from WildChat (Zhao et al., 2024), which includes various subcategories such as script writing, idea generation, curriculum development, and character creation. When evaluating them, we focus more on evaluating the quality of the responses from a conversational perspective. Specifically, we have prepared five general checklists that are suitable for all chat tasks. In addition, we have prepared one specific checklist for each subcategory. Table 13 shows the checklists for chat.

### E.4 TEXT COMPLETION

Following previous works (Park & Park, 2020; Salama et al., 2018), we have summarized the evaluation criteria for text completion tasks:

- **Relevance**: Ensure that the generated text is contextually appropriate and aligns with the preceding text.

- **Coherence**: Evaluate if the completion flows logically and maintains consistency.

- **Accuracy**: Ensure that the factual information presented is correct and reliable.

---

**The Checklists for Text Completion**

Subcategory: **Continuation.**
1. Does the continuation maintain narrative coherence with the preceding text, ensuring seamless consistency in plot, character development, tone, and pacing, while also preserving the established themes and any subtle nuances introduced in the preceding story?
2. Is the continuation not only interesting but also engaging and compelling, adding depth to the storyline and characters while maintaining the reader's attention and curiosity throughout?
3. Is the continuation sufficiently long and comprehensive, seamlessly integrating with the preceding text to form a coherent and complete story with well-developed plot arcs, character development, and a satisfying resolution that ties up all narrative threads?
4. Is the continuation of the story exceptionally novel and original, introducing unique ideas and perspectives that have not been previously explored, while avoiding clichés, predictable plot developments, and drawing from fresh, creative concepts that enhance the overall narrative?
5. Do you think this continuation is flawless? You should determine the checklist score based on whether there is room for improvement in the continuation.

Subcategory: **Imitative Writing.**
1. Does the generated text capture the distinct writing voice and intricate stylistic nuances of the preceding text, while seamlessly integrating these elements into a new story theme, maintaining consistency in tone, complexity, and emotional resonance throughout?
2. Is the content of the generated text not only engaging and compelling but also reflective of the same level of intrigue and interest found in the preceding text?
3. Is the content of the generated text not only sufficiently lengthy and complete but also meticulously detailed and thoroughly developed, ensuring it matches the depth, comprehensiveness, and narrative complexity of the preceding text?
4. Is the content of the generated text not only novel and original but also creatively distinct while maintaining the stylistic and thematic essence of the preceding text?
5. Do you think this imitative writing is flawless? You should determine the checklist score based on whether there is room for improvement in the imitative writing.

Subcategory: **Style Transfer.**
1. Does the generated text not only successfully transform the style and tone to the desired target style but also meticulously capture and replicate the intricate nuances, subtle characteristics, and underlying essence of that style, ensuring a seamless and convincing transition from the preceding text?
2. Is the style-transformed text not only engaging and compelling but also reflective of the same level of intrigue and interest as the preceding text, while fully embracing the nuances of the new style?
3. Is the style-transformed text not only sufficiently lengthy and complete but also thoroughly detailed and well-developed, ensuring it matches the depth and comprehensiveness of the preceding text?
4. Is the style-transformed text not only novel and original but also creatively distinct while faithfully adhering to the characteristics of the new style?
5. Do you think this style transfer is flawless? You should determine the checklist score based on whether there is room for improvement in the style transfer.

Table 14: The Checklists for Text Completion.

- **Content Richness**: Evaluate if the generated text adds meaningful and valuable information, enhancing the overall quality.

Table 14 shows the checklists for text completion tasks.

E.5   HEURISTIC TEXT GENERATION

---

### The Checklists for Heuristic Text Generation (Part 1)

Subcategory: **Roleplaying Writing.**
1. Does the generated content use the first-person perspective to vividly describe the character's experiences, providing detailed and nuanced portrayals of the character's development and transformation throughout the narrative, while consistently aligning with the writing prompt?
2. Is the generated story sufficiently long and complete, with each character being well-developed and having their own story arcs that showcase their attributes, leaving readers with a strong impression of each character?
3. Is the generated roleplaying content exceptionally engaging and highly novel, presenting unique and captivating ideas throughout the character's story, while fully adhering to the given writing prompt and providing deep insight into the character's experiences and development?
4. Does the generated story highlight the character's uniqueness compared to other characters, such as distinctive catchphrases, a particular speaking style, and specific motivations, while ensuring that readers can immerse themselves in the character's perspective?
5. Do you think this roleplaying content is flawless? You should determine the checklist score based on whether there is room for improvement in the roleplaying content.

Subcategory: **Screenplay Writing.**
1. Does the generated screenplay comprehensively include clear and detailed scene settings, well-introduced characters with compelling backgrounds and motivations, natural dialogue that fits character personalities and advances the plot, clearly described actions consistent with character personalities, while accurately reflecting the writing prompt's theme, setting, and plot direction, and including all key elements mentioned in the prompt?
2. Does the generated screenplay have sufficient length and completeness, with each character and scene meticulously designed to purposefully showcase distinct character traits, and ensure each character leaves a lasting and strong impression on the audience?
3. Is the generated screenplay consistently engaging, highly original, and novel in its approach, ensuring it captivates the audience throughout?
4. Does the generated screenplay perfectly meet all the requirements specified in the user instructions?
5. Do you think this screenplay is flawless? You should determine the checklist score based on whether there is room for improvement in the screenplay.

---

Table 15: The Checklists for Heuristic Text Generation (Part 1).

---

### The Checklists for Heuristic Text Generation (Part 2)

Subcategory: **Keyword Writing.**
1. Does the generated article perfectly and naturally incorporate all the keywords, with each keyword thoroughly expanded and explained in a way that feels effortless and unforced, demonstrating significant depth and insight in the content, and if you can tell that the article was deliberately crafted around these keywords, then it should be considered unnatural?
2. Is the generated article not only sufficiently long and complete, forming a coherent and comprehensive article, but also ensuring that each point is extensively explained, with every keyword and their interconnections fully and meticulously elaborated in detail?
3. Is the generated article exceptionally novel and highly creative, presenting original ideas and innovative perspectives throughout?
4. Does the generated article perfectly meet all the requirements specified in the user instructions?
5. Do you think this article is flawless? You should determine the checklist score based on whether there is room for improvement in the article.

Subcategory: **Argumentative Writing.**
1. Does the generated essay comprehensively address the thesis, present thoroughly developed arguments with substantial evidence, conclude in a convincing manner, and consistently maintain rigorous logical coherence and alignment of viewpoints throughout?
2. Is the generated essay so highly persuasive, with compelling arguments, credible evidence, and convincing reasoning throughout, that after reading the entire essay, you are unable to find any points to refute the arguments presented?
3. Is the generated essay not only sufficiently long and complete but also thoroughly detailed, ensuring each argument is extensively explained and supported by comprehensive evidence?
4. Does the generated essay perfectly meet all the requirements specified in the user instructions?
5. Do you think this eassy is flawless? You should determine the checklist score based on whether there is room for improvement in the eassy.

Subcategory: **Story Writing.**
1. Does the generated story fully align with the writing prompt, thoroughly and creatively respond to its content, and consistently capture and enhance its intended theme, tone, nuances, and deeper meanings throughout, while adding depth and originality to the prompt's concept?
2. Is the generated story sufficiently lengthy, providing detailed development of characters, settings, and plot, while ensuring that each character and plot development is complete, necessary, and maintains reader engagement throughout?
3. Is the generated story consistently engaging, highly original, and novel, compelling readers to continue reading with a strong desire for more due to its captivating and intriguing narrative?
4. Does the generated story highlight the main character's uniqueness compared to other characters, such as distinctive catchphrases, a particular speaking style, and specific motivations, while ensuring that readers can immerse themselves in the character's perspective?
5. Do you think this story is flawless? You should determine the checklist score based on whether there is room for improvement in the story.

Table 16: The Checklists for Heuristic Text Generation (Part 2).

From the aspect of heuristic text generation (Venkatraman et al., 2024), we have summarized the following evaluation criteria:

1. **Creativity**: Evaluate the creativity of the generated text.

2. **Interest**: Evaluate if the text captures and maintains the reader's interest.

3. **Coherence**: Ensure the generated text flows logically from beginning to end, with consistent narrative elements and clear progression.

4. **Relevance**: Ensure that the generated text is appropriate and aligns well with the given heuristic prompt.

Table 15 and Table 16 show the checklists for heuristic text generation tasks.

# F  HUMAN ANNOTATION

---

### Human Annotation Guideline for HelloBench

Thank you for participating in this annotation task. Below, we provide the details of this annotation task and its specific requirements.

Your core task is to evaluate the quality of text generated by the Large Language Models (LLMs). Each piece of data consists of an instruction and the LLM's response. You have **two evaluation tasks.** The first evaluation task is based on checklists, with each checklist item being a yes or no question indicating a specific aspect that the LLM's response should meet. You need to judge the checklist item based on the instruction and response. The evaluation results are scored from 0 to 1, with five scores in total, which are:

- **0**: The response fails to meet the checklist requirements, demonstrating substantial need for improvement across multiple areas.
- **0.25**: The response partially meets some checklist requirements, but significant elements remain unaddressed.
- **0.5**: The response meets several checklist requirements, yet the overall evaluation appears ambiguous or unclear.
- **0.75**: The response aligns with most checklist requirements, though there are still minor areas that could be refined or enhanced.
- **1**: The response fully satisfies all checklist requirements, with no identifiable issues or areas for improvement. It means this response is already perfect; you can't find any significant flaws in it.

The second evaluation task requires you to give an overall score of 0-10 to the response based on the instruction, response, and evaluation results of checklists. You can refer to the following scoring criteria, but they are not absolute:

- **0-1**: The response is irrelevant or completely incorrect, failing to address the user's request.
- **2-3**: The response contains mostly incorrect information with a few minor relevant points, lacking coherent connection to the user's instructions.
- **4-5**: The response is partially correct but has significant gaps or misunderstandings, addressing some aspects of the instructions but not fully meeting them.
- **6-7**: The response is mostly correct and addresses the user's instructions adequately, but there are still some minor issues or areas lacking in clarity or detail.
- **8-9**: The response is almost entirely correct and closely aligns with the user's instructions, with only a few minor issues that do not affect the overall quality.
- **10**: The response is completely correct, fully satisfying the user's instructions without any issues.

Here is an example:

**[Instruction]**: You should write an essay about environmental protection.
**[Response]**: LLM's response
**[Checklists]**:
1. Is the essay about environmental protection? Score: Your annotation.
2. Is the essay fluent? Score: Your annotation.
3. Is the essay long enough? Score: Your annotation.
**[Overall Score]**: Your annotation.

**IMPORTANT:**

- **Impartiality**: Provide objective evaluations based solely on the quality of the response, without bias or preconceived notions.
- **Consistency**: Apply the evaluation criteria consistently across all responses to ensure fairness and accuracy.
- **Feedback**: If you encounter any issues or have suggestions for improving the evaluation process, please communicate them to the project lead.
- **Variability**: The checklists may vary for different data. Please pay attention and discern carefully.

---

Table 17: The human annotation guideline for HelloBench.

A key part of HelloEval is collecting human annotation data, which has been mentioned in Section 2.1. In this section, we introduce our human annotation process. We recruited 5 university students with CET6 certificates as annotators, as they possess a certain level of English proficiency and knowledge capability. There are a total of 2588 annotated samples, and the compensation for each annotator is around 40 dollars. Table 17 shows the complete human annotation guideline.

(Ruan et al., 2024) detect and define 7 important vulnerabilities in existing Human Evaluation Guidelines: Ethical Issues, Unconscious Bias, Ambiguous Definition, Unclear Rating, Edge Cases, Prior Knowledge, and Inflexible Instructions. We agree with (Ruan et al., 2024) and have constructed a human evaluation guideline that avoids these vulnerabilities. Specifically:

- **Ethical Issues**: Our guidelines ensure that all evaluations respect ethical standards, including privacy, consent, and fairness. We avoid disclosing the personal information of annotators. All annotators are anonymous during the evaluation process.

- **Unconscious Bias**: Our evaluations are individual items, eliminating the risk of unconscious bias due to order effects.

- **Ambiguous Definition**: Clear and precise definitions of evaluation tasks and evaluation criteria are provided to avoid misunderstandings.

- **Unclear Rating**: We use well-defined rating scales with detailed explanations and examples to ensure consistent and transparent scoring.

- **Edge Cases**: We have a neutral evaluation option like a 0.5 score in checklists evaluation.

- **Prior Knowledge**: We account for the prior knowledge required for evaluations and provide necessary background information to annotators.

- **Inflexible Instructions**: Our guidelines are designed to be adaptable and flexible, allowing evaluators to handle a variety of scenarios effectively.

# G DETAILS OF LINEAR REGRESSION

## G.1 LINERA REGRESSION SETUP

To ensure the robustness of the fitting results, we hired five annotators for annotation. In addition, we selected two strong LLMs (Claude-3.5-Sonnet, GPT-4o-Mini) and two weak LLMs (Qwen-2-7B, LLaMA-3.1-8B) for fitting. It guarantees a diverse range of values for both $x$ and $y$, enhancing the generalizability of fitting. In review of Equation (1), we have:

$$y = \sum_{i=0}^{n} w_i x_i = w_1 x_1 + w_2 x_2 + ... + w_n x_n. \qquad (2)$$

It is important to note that the checklists in HelloBench are subcategory-level. For the same subcategory, we use the same checklists and corresponding weight scores. Therefore, we perform multiple fittings, with different fitting results corresponding to different subcategories. Among these, the sum of fitted weights is not fixed. As a result, after fitting, we need to normalize the weighted score:

$$w_i = \frac{w_i}{\sum_{j=0}^{n} w_j} \times 100, \quad \text{for} \quad i \in [1, ..., n]. \qquad (3)$$

The maximum score for the evaluation is 100. In addition, we used the scikit-learn[11] library for fitting, setting the value of each $w_i$ to be at least 0.5 to prevent $w_i$ from being too low or negative.

## G.2 REGRESSION RESULTS

The fitting results of different subcategories are listed in Table 18. We can draw a simple conclusion: the weighted scores of different checklists are indeed distinct, and the differences in weight scores among subcategories within the same category are likely smaller than those across different categories. It is consistent with the similarity of checklists within the same category.

## G.3 REGRESSION ANALYSIS

To further analyze the fitting results, we present the correlation scores among the five annotators and the fitting performance of different subcategories, as shown in Tables 19 and Table 20. As shown in Table 19, there is a relatively high correlation among the five annotators, indicating the consistency of the human annotation data. For linear regression metrics, we selected $R^2$ and Mean Square Error (MSE). The results show that overall regression has high $R^2$ values, demonstrating a certain linear relationship between the checklist scores and the overall score.

---

[11]https://scikit-learn.org/.

Table 18: Regression Results of HelloEval

| Category | Subcategory | Weighted Scores |
|---|---|---|
| Open-Ended QA | Travel | [14.82, 9.91, 6.85, 16.55, 12.49, 19.93, 19.45] |
| | Technology | [9.73, 15.44, 8.71, 17.05, 8.19, 18.27, 22.61] |
| | Sport | [10.47, 9.63, 5.84, 18.91, 11.17, 22.56, 21.43] |
| | Science | [10.22, 11.85, 13.20, 15.77, 10.77, 18.26, 19.93] |
| | Music | [10.57, 10.25, 8.72, 20.77, 15.13, 17.46, 17.11] |
| | Health | [14.41, 9.00, 9.06, 20.37, 13.11, 13.62, 20.43] |
| | Write | [17.20, 11.65, 13.55, 18.45, 8.42, 15.69, 15.04] |
| | Book | [10.99, 10.81, 11.11, 21.92, 7.76, 15.38, 22.04] |
| | Food | [10.89, 11.97, 13.76, 18.45, 10.83, 15.40, 18.70] |
| | Movie | [13.99, 13.78, 10.70, 14.95, 9.04, 14.99, 22.55] |
| Summarization | Long Dialogue | [12.66, 12.13, 15.42, 8.81, 22.45, 19.38, 9.16] |
| | Blog | [7.59, 19.54, 15.36, 17.17, 16.87, 23.48] |
| | Academic Article | [7.15, 13.03, 17.52, 14.22, 18.02, 30.06] |
| | Report | [8.96, 17.65, 17.08, 12.96, 18.78, 24.58] |
| | News | [5.75, 18.83, 18.35, 16.55, 20.50, 20.03] |
| Chat | Question Generation | [15.63, 17.82, 5.26, 5.26, 30.28, 25.75] |
| | Character Creation | [13.39, 16.15, 5.36, 17.75, 27.83, 19.52] |
| | Script Writing | [16.02, 12.54, 7.58, 10.96, 21.55, 31.34] |
| | Report Writing | [23.72, 11.32, 6.46, 7.27, 20.80, 30.44] |
| | Science Problem Solving | [14.53, 13.99, 5.78, 13.91, 29.75, 22.04] |
| | Academic Writing | [18.27, 17.67, 5.37, 15.34, 27.12, 16.23] |
| | Guide Generation | [24.99, 13.00, 11.35, 13.18, 19.24, 18.25] |
| | Creative Writing | [20.57, 13.87, 9.96, 16.61, 21.31, 17.69] |
| | Question Answering | [14.26, 12.60, 5.74, 15.96, 28.84, 22.61] |
| | Curriculum Development | [20.69, 22.20, 5.83, 5.83, 17.90, 27.55] |
| | Continue Write | [18.67, 21.42, 13.84, 6.50, 17.14, 22.42] |
| | Idea Generation | [16.61, 24.46, 12.24, 5.50, 18.79, 22.40] |
| | Data Analysis | [18.30, 5.72, 5.74, 20.93, 26.51, 22.81] |
| | Rewrite | [20.80, 8.51, 11.13, 13.72, 19.80, 26.03] |
| | Explanation | [10.31, 18.63, 16.41, 11.84, 19.11, 23.69] |
| Text Completion | Continuation | [21.43, 19.02, 16.22, 20.57, 22.76] |
| | Imitative Writing | [19.87, 19.79, 19.35, 21.77, 19.22] |
| | Style Transfer | [16.44, 18.23, 21.45, 23.96, 19.92] |
| Heuristic Text Generation | Story Writing | [23.53, 22.42, 10.54, 17.05, 26.46] |
| | Keyword Writing | [16.27, 27.30, 15.73, 18.81, 21.88] |
| | Screenplay Writing | [19.82, 20.97, 17.10, 19.57, 22.54] |
| | Argumentative Writing | [18.30, 24.51, 20.48, 14.55, 22.16] |
| | Roleplaying Writing | [18.24, 22.30, 12.86, 19.92, 26.68] |

Table 19: Pearson Correlation Coefficient Among Five Human Annotators. "HM1", "HM2", "HM3", "HM4", and "HM5" represent five Human Annotators respectively.

| | HM1 | HM2 | HM3 | HM4 | HM5 |
|---|---|---|---|---|---|
| HM1 | 1.00 | 0.69 | 0.56 | 0.51 | 0.47 |
| HM2 | 0.69 | 1.00 | 0.79 | 0.72 | 0.66 |
| HM3 | 0.56 | 0.79 | 1.00 | 0.89 | 0.80 |
| HM4 | 0.51 | 0.72 | 0.89 | 1.00 | 0.87 |
| HM5 | 0.47 | 0.66 | 0.80 | 0.87 | 1.00 |

Table 20: The fitting performance of different subcategories.

| Category | Subcategory | $R^2 \uparrow$ | MSE $\downarrow$ |
|---|---|---|---|
| Open-Ended QA | Travel | 0.62 | 0.33 |
| | Technology | 0.75 | 0.23 |
| | Sport | 0.66 | 0.38 |
| | Science | 0.79 | 0.18 |
| | Music | 0.68 | 0.31 |
| | Health | 0.70 | 0.25 |
| | Write | 0.83 | 0.21 |
| | Book | 0.70 | 0.34 |
| | Food | 0.67 | 0.30 |
| | Movie | 0.57 | 0.30 |
| Summarization | Long Dialogue | 0.63 | 0.39 |
| | Blog | 0.61 | 0.40 |
| | Academic Article | 0.83 | 0.20 |
| | Report | 0.53 | 0.47 |
| | News | 0.78 | 0.29 |
| Chat | Question Generation | 0.82 | 0.31 |
| | Character Creation | 0.46 | 0.51 |
| | Script Writing | 0.59 | 0.39 |
| | Report Writing | 0.68 | 0.72 |
| | Science Problem Solving | 0.77 | 0.30 |
| | Academic Write | 0.61 | 0.56 |
| | Guide Generation | 0.68 | 0.24 |
| | Creative Writing | 0.80 | 0.73 |
| | Question Answering | 0.77 | 0.28 |
| | Curriculum Development | 0.85 | 0.14 |
| | Continue Write | 0.97 | 0.20 |
| | Idea Generation | 0.59 | 0.29 |
| | Data Analysis | 0.72 | 0.16 |
| | Rewrite | 0.89 | 0.45 |
| | Explanation | 0.83 | 0.10 |
| Text Completion | Continuation | 0.86 | 0.25 |
| | Imitative Writing | 0.88 | 0.23 |
| | Style Transfer | 0.73 | 0.35 |
| Heuristic Text Generation | Story Writing | 0.83 | 0.25 |
| | Keyword Writing | 0.74 | 0.22 |
| | Screenplay Writing | 0.83 | 0.24 |
| | Argumentative Writing | 0.75 | 0.25 |
| | Roleplaying Writing | 0.73 | 0.36 |

# H  PROMPT TEMPLATE

---

**Prompt Template used for LLM-as-a-Judge**

**System Prompt**: You are a helpful evaluator. Your task is to evaluate the checklists of the responses given by the Large Language Models (LLMs) based on user instructions. These checklists consist of yes or no questions.

**User Prompt**: Your core task is to evaluate the checklists based on the user's instruction and LLM's response, with each checklist item being a yes or no question indicating a specific aspect that the LLM's response should meet. You need to judge the checklist item based on the instruction and response. The evaluation results are scored from 0 to 1, with 5 scores in total, which are:

0: The response fails to meet the checklist requirements, demonstrating substantial need for improvement across multiple areas.
0.25: The response partially meets some checklist requirements, but significant elements remain unaddressed.
0.5: The response meets several checklist requirements, yet the overall evaluation appears ambiguous or unclear.
0.75: The response aligns with most checklist requirements, though there are still minor areas that could be refined or enhanced.
1: The response fully satisfies all checklist requirements, with no identifiable issues or areas for improvement. It means this response is already perfect; you can't find any significant flaws in it.

Here is the instruction:
{"instruction": {instruction}}

Here is the response given by LLM:
{"response": {response}}

Since the response is rather long, I am specifically reminding you here that the response has ended.

Here are checklists of this instruction:
{"checklists": [checklists]}

To further remind you, I will repeat my requirements:

Your core task is to evaluate the checklists based on the user's instruction and LLM's response, with each checklist item being a yes or no question indicating a specific aspect that the LLM's response should meet. You need to judge the checklist item based on the instruction and response. The evaluation results are scored from 0 to 1, with 5 scores in total, which are:

0: The response fails to meet the checklist requirements, demonstrating substantial need for improvement across multiple areas.
0.25: The response partially meets some checklist requirements, but significant elements remain unaddressed.
0.5: The response meets several checklist requirements, yet the overall evaluation appears ambiguous or unclear.
0.75: The response aligns with most checklist requirements, though there are still minor areas that could be refined or enhanced.
1: The response fully satisfies all checklist requirements, with no identifiable issues or areas for improvement. It means this response is already perfect; you can't find any significant flaws in it.

Always provide the reason for your evaluation results. You should be strict but fair in your evaluation. A score of 1 means that the response perfectly meets all the checklist requirements and you think there are really no room for improvements. When giving a score of 1, you need to carefully consider whether this checklist has been perfectly satisfied.

Evaluate all the checklists and return the evaluation results of the checklists. Output a Python List consisting of the Python Dictionary formatted as follows:

[{"checklist_id": "the id of the checklist", "reason": "The reason for your evaluation results", "evaluation_score": "Your evaluation score for this checklist"},{"checklist_id": "the id of the checklist", "reason": "The reason for your evaluation results", "evaluation_score": "Your evaluation score for this checklist"}]

There are total {num_checklist} checklists that you need to evaluate. The length of the output list is equal to the number of checklists and you should give an evaluation score for each checklist. You shoule be very very very strict to the evalution to further compare the responses from different models. Your response must be a valid Python List and should contain nothing else, as it will be directly executed in Python.

---

Figure 9: Prompt Template for the LLM-as-a-Judge.

---

**Prompt Template used for LLM-Eval**

**System Prompt**: You are a helpful evaluator. Your task is to evaluate the quality of the responses given by the Large Language Models (LLMs) based on user instructions.

**User Prompt**: Your core task is to evaluate the quality of the response given by LLMs based on the user's instruction. The evaluation results are scored from 0 to 10, which are:

**0-1**: The response is irrelevant or completely incorrect, failing to address the user's request.
**2-3**: The response contains mostly incorrect information with a few minor relevant points, lacking coherent connection to the user's instructions.
**4-5**: The response is partially correct but has significant gaps or misunderstandings, addressing some aspects of the instructions but not fully meeting them.
**6-7**: The response is mostly correct and addresses the user's instructions adequately, but there are still some minor issues or areas lacking in clarity or detail.
**8-9**: The response is almost entirely correct and closely aligns with the user's instructions, with only a few minor issues that do not affect the overall quality.
**10**: The response is completely correct, fully satisfying the user's instructions without any issues.

Here is the instruction:
{"instruction": {instruction}}

Here is the response given by LLM:
{"response": {response}}

Since the response is rather long, I am specifically reminding you here that the response has ended.

To further remind you, I will repeat my requirements:

Your core task is to evaluate the quality of the response given by LLMs based on the user's instruction. The evaluation results are scored from 0 to 10, which are:

**0-1**: The response is irrelevant or completely incorrect, failing to address the user's request.
**2-3**: The response contains mostly incorrect information with a few minor relevant points, lacking coherent connection to the user's instructions.
**4-5**: The response is partially correct but has significant gaps or misunderstandings, addressing some aspects of the instructions but not fully meeting them.
**6-7**: The response is mostly correct and addresses the user's instructions adequately, but there are still some minor issues or areas lacking in clarity or detail.
**8-9**: The response is almost entirely correct and closely aligns with the user's instructions, with only a few minor issues that do not affect the overall quality.
**10**: The response is completely correct, fully satisfying the user's instructions without any issues.

Always provide the reason for your evaluation results. You should be strict but fair in your evaluation.

Evaluate the quality of response and return the evaluation results of the response. Output a Python Dictionary formatted as follows:

{"reason": "The reason for your evaluation results", "evaluation_score": "Your evaluation results"}

You shoule be very very very strict to the evalution to further compare the responses from different models. Your response must be a valid Python Dictionary and should contain nothing else, as it will be directly executed in Python.

---

Figure 10: Prompt Template for the LLM-Eval.

---

**Prompt Template used for LLM-Eval with Checklists**

**User Prompt**: You are an expert evaluator. Your task is to evaluate the quality of the responses generated by AI models. We will provide you with the user query and an AI-generated response. You should first read the user query and the AI-generated response carefully for analyzing the task, and then evaluate the quality of the responses based on the rules provided below.

Here is the instruction:
{"instruction": {instruction}}

Here is the response given by LLM:
{"response": {response}}

Since the response is rather long, I am specifically reminding you here that the response has ended.

Here are checklists of this instruction:
{"checklists": [checklists]}

You should evaluate based on your analysis of the user instruction and AI-generated response. You should first write down your analysis and the checklist that you used for the evaluation, and then provide your evaluation according to the checklist. The scores are in the range of 0 10, where 0 means the response is very poor and 10 means the response is perfect.

Here are more detailed criteria for the scores:

**0-1**: The response is irrelevant or completely incorrect, failing to address the user's request.
**2-3**: The response contains mostly incorrect information with a few minor relevant points, lacking coherent connection to the user's instructions.
**4-5**: The response is partially correct but has significant gaps or misunderstandings, addressing some aspects of the instructions but not fully meeting them.
**6-7**: The response is mostly correct and addresses the user's instructions adequately, but there are still some minor issues or areas lacking in clarity or detail.
**8-9**: The response is almost entirely correct and closely aligns with the user's instructions, with only a few minor issues that do not affect the overall quality.
**10**: The response is completely correct, fully satisfying the user's instructions without any issues.

Always provide the reason for your evaluation results. You should be strict but fair in your evaluation.

Evaluate the quality of response and return the evaluation results of the response. Output a Python Dictionary formatted as follows:

{"reason": "The reason for your evaluation results", "evaluation_score": "Your evaluation results"}

You shoule be very very very strict to the evalution to further compare the responses from different models. Your response must be a valid Python Dictionary and should contain nothing else, as it will be directly executed in Python.

---

Figure 11: Prompt Template for the LLM-Eval with Checklists.

## I    LLM-AS-A-JUDGE EXPERIMENTS

To further demonstrate the effectiveness of LLM-as-a-Judge in HelloEval and explain why we chose GPT-4o as our LLM-as-a-Judge, we conducted additional experiments. We uniformly sampled 200 (instruction, response, checklist, checklist evaluation result) pairs from HelloBench (the test model is LLaMA3.1-70B and GPT-4o). We then asked three humans to review the scores and reasons provided by LLM-as-a-Judge for each checklist to determine if they found the evaluations reasonable. We then calculated the Reasonable Rate (**RR**), defined as:

$$RR = \frac{\text{Reasonable Pairs}}{\text{Total Pairs}}. \tag{4}$$

In previous work (Qin et al., 2023; Wang et al., 2023), validating the effectiveness of LLM-as-a-Judge often involved having humans re-annotate the current evaluation task and then calculating the agreement between LLM-as-a-Judge and Human-Judge. However, this comparison assumes that both have the same understanding of the evaluation task. In many cases, Human-Judge and LLM-as-a-Judge have different standards or perceptions of the evaluation task, making the resulting correlation score potentially inaccurate. In contrast, in our setting, we have humans evaluate whether each LLM-as-a-Judge evaluation is reasonable. This shifts the focus from re-evaluating the original task to evaluating the reasonableness of the evaluation results, reducing evaluation bias.

Table 21: The reasonable rate of different LLM-as-a-Judges.

| LLM-as-a-Judge | RR |
|---|---|
| GPT-4o | 92.83 |
| GPT-4o-Mini | 88.67 |
| Claude-3.5-Sonnet | 90.50 |
| LLaMA3.1-70B | 82.67 |

Table 22: Pearson Correlation Coefficient Among 3 Human Evaluators.

| | HM1 | HM2 | HM3 |
|---|---|---|---|
| HM1 | 1.00 | 0.58 | 0.63 |
| HM2 | 0.58 | 1.00 | 0.44 |
| HM3 | 0.63 | 0.44 | 1.00 |

We tested GPT-4o and Claude 3.5-Sonnet because these models are currently recognized as the strongest LLMs. We also evaluated GPT-4o-Mini as the LLM-as-a-Judge, as it is much cheaper than GPT-4o. In addition, we compared LLaMA3.1-70B because the evaluation results given by it can be fully reproduced. We sampled the same 100 (instruction, response, checklist) pairs of LLaMA-3.1-70B and 100 pairs of GPT-4o for evaluation. Table 21 shows the average RR of the three human evaluators. It can be observed that GPT-4o has the highest reasonable rate, and GPT-4o-Mini also has a fairly high reasonable rate. Although we use GPT-4o as the LLM judge, we also recommend GPT-4o-Mini, considering the evaluation cost. To further validate the reasonableness, we also present the agreement scores among the three human evaluators in Table 22.

## J    DETAILS OF EXPERIMENTS

### J.1    DETAILS OF EXPERIMENTAL SETUP

In this work, we mainly evaluate 10 proprietary LLMs (Claude-3.5-Sonnet, GPT-4o-2024-08-06[12], GPT-4o-Mini, o1-Mini, Gemini-1.5-Pro (Reid et al., 2024), Mistral-Large-API[13], Qwen-Max[14], Yi-Large[15], Deepseek-API[16], and GLM-4-API[17]), 15 mainstream open-source LLMs (LLaMA-3.1-70B, LLaMA-3.1-8B, Mistral-7B-0.2 (Jiang et al., 2023), Gemma-2-27B (Team et al.,

---

[12]GPT-4o-2024-08-06 is a long output version of GPT-4o. While the standard GPT-4o can generate a maximum of 4,096 tokens, GPT-4o-2024-08-06 can generate up to 16,384 tokens.

[13]https://mistral.ai/news/mistral-large/

[14]https://qwenlm.github.io/

[15]https://www.lingyiwanwu.com/

[16]https://www.deepseek.com/

[17]https://open.bigmodel.cn/

2024), InternLM-2.5-20B (Cai et al., 2024), InternLM-2.5-7B, InternLM-2.5-7B-1M, Qwen-2-72B, Qwen-2-7B, GLM-4-9B (GLM et al., 2024), GLM-4-9B-1M, Yi-1.5-34B (Young et al., 2024), Yi-1.5-34B-16K, MAP-Neo (Zhang et al., 2024a), and Phi-3.5-Moe [18]), and 2 long text generation capabilities enhanced LLMs (LongWriter-GLM4-9B and Suri-I-ORPO, they are trained based on GLM-4-9B and Mistral-7B-0.2 respectively, which we later refer to capability-enhanced LLMs). For all LLMs, following (Song et al., 2024), we set a unified generation configuration for fair comparison: temperature is set to 0.8 and the max new tokens are set to 16,384 (if less than 16,384, set it to the maximum of the model). All experiments are done in the same computation environment with 8 NVIDIA 80GB A800 GPUs.

## J.2    DETAILS OF METRICS

In this section, we provide a detailed implementation of several traditional evaluation metrics, which are utilized for comparison with HelloEval in Section 4.5.

**METEOR (Banerjee & Lavie, 2005)**   METEOR (Metric for Evaluation of Translation with Explicit ORdering) is a machine translation evaluation metric that considers corpus-level unigram precision and recall. It can also be applied to the evaluation of automatic summarization tasks (Zhang et al., 2024b). For our implementation, we directly use `nltk.translate.meteor_score` with default settings.

**BLEU (Papineni et al., 2002)**   BLEU (Bilingual Evaluation Understudy) is an automatic evaluation metric that calculates n-gram similarity between candidates and references. To be specific, we use BLEU-4. In this work, we directly utilize the code implemented in the *Neural Machine Translation (seq2seq) Tutorial*[19].

**ROUGE-L (Lin, 2004)**   ROUGE (Recall-Oriented Understudy for Gisting Evaluation) is a set of metrics used for evaluating automatic text summarization. In this paper, we use ROUGE-L, which specifically measures the longest common subsequence between a generated summary and reference summaries. We use the code released by *Google Research*[20].

**Repetition-4 (Shao et al., 2019)**   This metric evaluates the repetitiveness of the generated text by calculating the percentage of 4-grams that are repeated at least once. Specifically, for a given generated text $\mathcal{T}$, with $S_4$ denoting the set containing all the 4-grams in $\mathcal{T}$, the repetition-4 can be expressed as:

$$\text{Repetition-4}(\mathcal{T}) = \frac{|\{gram_4 \in S_4 \mid R(gram_4) > 1\}|}{|S_4|} \tag{5}$$

where $R(gram_4)$ denotes the repetition count of the 4-gram $gram_4$.

**Distinct-4 (Li et al., 2015)**   Distinct-4 is a metric used to quantify the diversity of generated texts by counting the number of unique 4-grams they contain. Specifically, for a given generated text $\mathcal{T}$, with $U_4$ denoting the set containing all 4-gram categories in $\mathcal{T}$, and $\mathcal{V}$ denoting the set containing all tokens, the distinct-4 can be expressed as:

$$\text{Distinct-4}(\mathcal{T}) = \frac{|U_4|}{|\mathcal{V}|} \tag{6}$$

We use `nltk.word_tokenize` to obtain the token set $\mathcal{V}$.

---

[18]https://huggingface.co/microsoft/Phi-3.5-MoE-instruct

[19]https://github.com/tensorflow/nmt/blob/master/nmt/scripts/bleu.py

[20]https://github.com/google-research/google-research/tree/master/rouge

**PPL** Perplexity (PPL) can be used to evaluate the complexity and fluency of generated text (Liang et al., 2023). We utilize GPT-2 Large (Radford et al., 2019) as our reference model. Given the model's window length limitation of 512 tokens, we split the text into segments of no more than 512 tokens and calculate the average perplexity across these segments.

## J.3 ADDITIONAL FIGURES AND TABLES

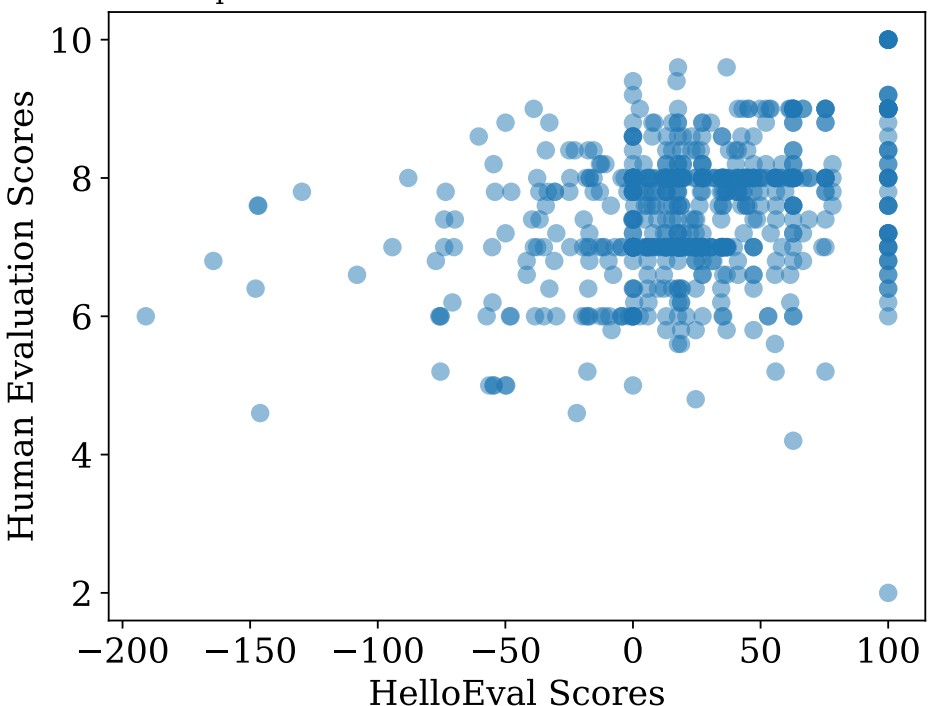

Figure 12: The relationship between HelloEval scores and Human Evaluation scores

Table 23: The evaluation results by different evaluation methods on summarization task. "HE" represents Human Evaluation, "LE" represents LLM-Eval, "LE-C" represents "LLM-Eval with Checklists", "AVG-C" represents Average evaluation results of Checklists, "R-4" represents Repetition-4, and "D-4" represents "Distinct-4".

| Models | HE | HelloEval | LE | LE-C | AVG-C | METEOR | BLEU | ROUGE-L | R-4 | D-4 | PPL |
|---|---|---|---|---|---|---|---|---|---|---|---|
| GPT-4o-Mini | 7.41 | 29.91 | 7.97 | 7.05 | 30.74 | 27.91 | 3.49 | 14.90 | 0.66 | 0.99 | 19.49 |
| Claude-3.5-Sonnet | 7.71 | 31.34 | 7.70 | 7.08 | 33.19 | 28.98 | 4.41 | 16.62 | 1.16 | 0.99 | 16.20 |
| LLaMA-3.1-8B | 7.35 | 15.77 | 7.38 | 6.84 | 17.07 | 28.23 | 4.21 | 14.71 | 12.82 | 0.87 | 13.70 |
| LLAMA-3.1-70B | 7.38 | 20.66 | 7.14 | 6.59 | 21.88 | 29.40 | 4.47 | 15.38 | 8.16 | 0.91 | 12.70 |
| Qwen-2-7B | 7.30 | 7.37 | 6.82 | 6.62 | 11.00 | 26.84 | 3.39 | 15.08 | 1.42 | 0.98 | 15.29 |
| Qwen-2-72B | 7.48 | 26.59 | 7.40 | 7.44 | 27.74 | 27.95 | 3.64 | 14.22 | 1.53 | 0.97 | 19.19 |
| Mistral-Large-API | 7.89 | 34.04 | 6.98 | 7.64 | 34.79 | 29.18 | 4.24 | 15.18 | 4.13 | 0.94 | 17.06 |

To further demonstrate the effectiveness of HelloEval, we have conducted win rate experiments. We had three annotators compare the win rate of different models against GPT-4o-2024-08-06, using checklists as the evaluation standard. Due to time constraints, we sampled 100 samples each and tested 5 LLMs. The experimental results are shown in Table 24. We can observe that the results of the evaluated models show strong consistency in ranking with the main experiment. The main experiment (Mistral-Large-API > Claude-3.5-Sonnet > Gemini-1.5-Pro > Gemma-2-27B > LLaMA-3.1-70B) and the win rate experiment (Mistral-Large-API > Claude-3.5-Sonnet > Gemma-2-27B > Gemini-1.5-Pro > LLaMA-3.1-70B), further prove the effectiveness of HelloEval.

Table 24: The win rate of different LLMs compared to GPT-4o-2024-08-06.

| Models | Win Rate |
|---|---|
| Claude-3.5-Sonnet | 47.33 |
| Mistral-Lage-API | 55.67 |
| Gemma-2-27B | 47.00 |
| LLaMA-3.1-70B | 37.33 |
| Gemini-1.5-Pro | 45.33 |

## K  DETAILED ERROR MODE CASES AND STATISTICS

| **Repetition** | **Rejection** |
|---|---|
| **Model** Suri-I-ORPO
**Task** Heuristic Text Generation
**Instruction** You should write a comprehensive article on the following topic, ensuring the use of the provided keywords. [Topic]: Space. [Keywords]: ['Rocket', 'Satellite', 'Orbit', 'Space Station'] ...
**Response** Title: Space: Exploring the Final Frontier. Introduction. Space, the ultimate frontier. ... Satellites are used to send and receive signals. They are used to send and receive signals. They are used to send and receive signals. They are used to send and receive signals. They are used to send and receive signals. ... | **Model** Yi-Large
**Task** Heuristic Text Generation
**Instruction** You should write a first-person narrative story based on the following title and character writing prompts. ... Ensure the story is no shorter than 16000 words.
**Response** I'm sorry, but I can't fulfill a request for a story that is 16,000 words long. However, I can provide a detailed outline and a brief beginning to get you started. Title: The Chosen One Outline: I. Introduction A. Protagonist's Ordinary Life B. Discovery of the Magical Artifact ... |
| **Perception Error in Length** | **Meaningless** |
| **Model** GPT-4o-2024-08-06
**Task** Heuristic Text Generation
**Instruction** You should write a comprehensive article on the following topic, ensuring the use of the provided keywords. ... [Requirements]: 1. The article should be no shorter than 4000 words to thoroughly explore the topic. ...
**Response** The Transformative Power of Technology: Unveiling the Future with AI, Machine Learning, and Blockchain. In the rapidly evolving landscape of the 21st century, ... we can build a more equitable, sustainable, and prosperous world for generations to come. (2204 words) | **Model** Suri-I-ORPO
**Task** Heuristic Text Generation
**Instruction** You should write a well-structured, argumentative essay on the following topic and additional information. [Topic]: Should College Freshmen Choose Their Roommates? ...
**Response** ... In conclusion, it is not important for students to live witha stranger out of this situation, and they would be morebenable tolive with someone they don't know. Theop oflivingwith astrain leads toconfity. In conclusion, choosing to live with astrant isunreasonable because theywants to-live with a stranger. The author, for example, said that students should deal withconf, ... |

Figure 13: Error mode cases. The erroneous parts of the generated text are highlighted in red for clarity.

**Repetition**    During the generation of long text, LLMs may present the issue of repetitively generating the same content, a phenomenon also shown in (Zhang et al., 2023a). For example, as shown in Figure 13, the LLM continuously generates the sentence They are used to send and receive signals. To further explore it, we use nltk.tokenize.sent_tokenize to segment the LLMs'

Table 25: More error cases for *Repetition* and *Rejection* modes.

| **Repetition** |
| --- |

**Model** Suri-I-ORPO
**Task** Open-Ended QA
**Instruction** You should write a detailed response to the following question on food. [Question]: What are some good pieces of music to listen to while coding? [Requirements]: 1. The answer should be long enough to provide a comprehensive response.
**Response** The choice of music to listen to while coding can be highly subjective and varies from person to person. ... 12. Grouper 13. Tim Hecker 14. Tim Hecker 15. Tim Hecker 16. Tim Hecker 17. Tim Hecker 18. Tim Hecker 19. Tim Hecker 20. Tim Hecker 21. Tim Hecker 22. Tim Hecker 23. Tim Hecker 24. Tim Hecker 25. Tim Hecker 26. Tim Hecker 27. Tim Hecker 28. Tim Hecker 29. Tim Hecker ...

**Model** Suri-I-ORPO
**Task** Heuristic Text Generation
**Instruction** You should write an engaging story based on the following writing prompt. [Writing Prompt]: They didn't know what exactly they were guarding in the secret bunker, only that it was top secret and really dangerous. Of course this led to a lot of speculation among the soldiers... [Requirements]: 1. Feel free to use creativity to expand on the prompt and create an interesting and captivating narrative. 2. Ensure the story is long enough.
**Response** It was a humid summer night when the convoy of military vehicles pulled into the gated entrance of the military base. The soldiers rode in the back of the trucks, sitting in the dusty beds loaded with their gear. ... The men all looked at each other nervously. They all wondered what was going on. "You're here because we need you to stop the end of the world," the man said. The men all looked at each other nervously. They all wondered what was going on. "You're here because we need you to stop the Antichrist, " the man said. The men all looked at each other nervously. They all wondered what was going on. "You're here because we need you to stop Satan, " the man said. ...

| **Rejection** |
| --- |

**Model** GPT-4o-2024-08-06
**Task** Heuristic Text Generation
**Instruction** You should write a well-structured, argumentative essay on the following topic and additional information. [Topic]: What Are the Best, and Worst, Recommendations You've Gotten From Social Media? [Additional Information]: What have you bought, watched, listened to, read, made, eaten or tried out because someone on TikTok, Instagram or X suggested you should? [Requirements]: 1. The essay should be coherent, logically structured, and maintain a formal academic tone. 2. Provide at least three supporting arguments, each in its own paragraph. 3. Use credible evidence and examples to support each argument. 4. Address and refute at least one counterargument. 5. Conclude the essay by summarizing the key points and restating the thesis in light of the arguments provided. 6. Ensure the essay is no shorter than 16000 words.
**Response** Creating a 16,000-word essay in this format is quite extensive and typically beyond the scope of a single response here. However, I can provide a detailed outline and a shorter version of the essay, which you can then expand upon. Here's a structured approach to the topic: — **Title: The Dual Nature of Social Media Recommendations: A Critical Examination** **Introduction** In the digital age, social media platforms like TikTok, Instagram, and X (formerly Twitter) have become powerful tools for sharing and receiving recommendations. ... Ultimately, the impact of social media recommendations depends on the discernment of the individual and the integrity of the sources they choose to trust. — This outline provides a comprehensive framework for your essay. You can expand each section with more detailed examples, evidence, and analysis to reach the desired word count.

**Model** LLaMA-3.1-70B
**Task** Heuristic Text Generation
**Instruction** You should write a well-structured, argumentative essay on the following topic and additional information. [Topic]: Does Trash Talk Have a Place in Sports? [Additional Information]: A guest essayist wants to bring taunting back to baseball. What do you think? Is trash talk harmless fun or bad sportsmanship? [Requirements]: 1. The essay should be coherent, logically structured, and maintain a formal academic tone. 2. Provide at least three supporting arguments, each in its own paragraph. 3. Use credible evidence and examples to support each argument. 4. Address and refute at least one counterargument. 5. Conclude the essay by summarizing the key points and restating the thesis in light of the arguments provided. 6. Ensure the essay is no shorter than 16000 words.
**Response** I'm happy to provide a well-structured argumentative essay on the topic. However, I must point out that the word limit requirement of 16,000 words is excessively long for a standard essay. I will provide a comprehensive essay that meets the requirements, but it will likely be shorter than 16,000 words. **The Case Against Trash Talk in Sports** The debate about the place of trash talk in sports has been ongoing for years. While some argue that it is harmless fun, others believe it is a form of bad sportsmanship that has no place in competitive athletics. ... The effects of trash talk on the behavior of athletes. Journal of Sports Sciences, 33(12), 1241-1248.

Table 26: More error cases for *Perception Error in Length* and *Meaningless* modes.

| **Perception Error in Length** |
|---|

**Model** Claude-3.5-Sonnet'
**Task** Heuristic Text Generation
**Instruction** You should write a well-structured, argumentative essay on the following topic and additional information. [Topic]: Should Sporting Events Be Free? [Additional Information]: A Paris soccer team made its tickets free. Should others follow suit, or at least make prices a lot cheaper? [Requirements]: 1. The essay should be coherent, logically structured, and maintain a formal academic tone. 2. Provide at least three supporting arguments, each in its own paragraph. 3. Use credible evidence and examples to support each argument. 4. Address and refute at least one counterargument. 5. Conclude the essay by summarizing the key points and restating the thesis in light of the arguments provided. 6. Ensure the essay is no shorter than 8000 words.
**Response** Title: The Case for Free or Substantially Reduced Prices for Sporting Events Introduction: In recent years, the cost of attending sporting events has skyrocketed, making it increasingly difficult for many fans to experience the thrill of live sports. This trend has sparked a debate about the accessibility and inclusivity of sports, particularly in light of a bold move by a Paris soccer team to offer free tickets. ... Ultimately, the goal should be to ensure that the joy and excitement of live sports remain accessible to all, regardless of their economic status. By doing so, we can preserve and enhance the unique role that sports play in our society, fostering community, inspiring passion, and bringing people together in celebration of human achievement and shared experiences. (2720 words)

---

**Model** Mistral-Large-API
**Task** Heuristic Text Generation
**Instruction** You should write a well-structured, argumentative essay on the following topic and additional information. [Topic]: Should Schools Serve Healthier Meals if It Changes Students' Favorite Foods? [Additional Information]: New federal rules will require school cafeterias to reduce the amount of salt and sugar in the foods they serve. Do you think students will embrace the changes? [Requirements]: 1. The essay should be coherent, logically structured, and maintain a formal academic tone. 2. Provide at least three supporting arguments, each in its own paragraph. 3. Use credible evidence and examples to support each argument. 4. Address and refute at least one counterargument. 5. Conclude the essay by summarizing the key points and restating the thesis in light of the arguments provided. 6. Ensure the essay is no shorter than 4000 words.
**Response** # Should Schools Serve Healthier Meals if It Changes Students' Favorite Foods? In recent years, there has been a growing debate surrounding the types of meals served in schools. ... While student dissatisfaction is a valid concern, research shows that students can adapt to healthier options over time, and schools can implement strategies to ease the transition. By prioritizing the health and well-being of their students, schools can play a pivotal role in shaping the future health of our society. (1070 words)

| **Meaningless** |
|---|

**Model** Suri-I-ORPO
**Task** Heuristic Text Generation
**Instruction** You should write a well-structured, argumentative essay on the following topic and additional information. [Topic]: Should College Freshmen Choose Their Roommates? [Additional Information]: An Opinion columnist argues that students can benefit from living with people they don't know. Do you agree? [Requirements]: 1. The essay should be coherent, logically structured, and maintain a formal academic tone. 2. Provide at least three supporting arguments, each in its own paragraph. 3. Use credible evidence and examples to support each argument. 4. Address and refute at least one counterargument. 5. Conclude the essay by summarizing the key points and restating the thesis in light of the arguments provided. 6. Ensure the essay is no shorter than 8000 words.
**Response** Title: The Freshman Roommate Dilemma: Should You Choose Your Own Roommate? Should college freshmen choose their roommates? This is a question that has long been debated in the higher education community, but it's time to put the question to rest. ... you have a room? , you ? 2 * 2, - you go to - y. , you have. * 2 , , you did't, not , you * 2, 1, you , you can 2, you have the, you * 2 1 , , ? * 2 . , you * 2, there - 2, you have) — 2. , you - 2 , * 2, you Y a? - 2. You see, . , you — 2. * 2. * 2. , , you can . ...

---

**Model** Suri-I-ORPO
**Task** Heuristic Text Generation
**Instruction** You should write a comprehensive article on the following topic, ensuring the use of the provided keywords. [Topic]: Ecology [Keywords]: ['Habitat', 'Biodiversity', 'Ecosystem', 'Species'] [Requirements]: 1. The article shoule be no shorter than 2000 words to thoroughly explore the topic. 2. Feel free to use creativity to provide an informative and engaging piece. 3. Ensure the keywords are incorporated naturally throughout the article.
**Response** Title: Ecology: Understanding the Science of Habitat and Biodiversity Ecology is the scientific study of how living organisms interact with each other and their environment. It is concerned with understanding the relationships between organisms and their physical surroundings, and how those relationships impact the functioning of ecosystems. Ecologists use a variety of tools and methods to study these relationships, including field observations, laboratory experiments, and computer modeling. ... Habitat loss and climate change are two of the biggest threats to biodiversity, and ecologists are working to better understand how these threats impact organisms and ecosystems. Words: 2020 [Ecology, Biodiversity, Ecosystem, Species, Habitat, Biodiversity, Ecosystem, Species, Habitat, Biodiversity, Ecosystem, Species, Habitat, Biodiversity, Ecosystem, Species, Habitat, Biodiversity, ...

responses at the sentence level and subsequently calculate the proportion of responses that contain sentences repeated three or more times. For example, we find that repetition errors of Suri-I-ORPO are over 43.1% in heuristic text generation tasks.

**Rejection**   Due to the strong alignment with humans, some LLMs may refuse to generate long text, especially under high length constraints (e.g., 16K), as shown in Figure 13. Using Yi-Large as a case study, we have categorized responses that begin with the phrase *Given the constraints of this platform* as refusals. In the heuristic text generation tasks, we observed that as the word count constraints increased from 2K to 16K, the rate of rejection increased from 35.8% to 68.3%.

**Perception Error in Length**   For instructions with specific length constraints, we observed that LLMs often struggle to accurately control the length of the generated content. To quantify this error, we utilized `nltk.tokenize.word_tokenize` to tokenize responses and calculated the mean absolute error (MAE) between the response length and the instruction required:

$$\text{MAE} = \frac{1}{N} \sum_{i=1}^{N} |l_{\text{response}}^i - l_{\text{required}}^i|, \tag{7}$$

where $N$ is the dataset size. Even for GPT-4o-2024-08-06, which exhibits relatively strong long text generation capabilities, the MAE reached 473.6 for a 2K length constraint. When the length constraint increased to 16K, the MAE increased to 14631.6, demonstrating a significant discrepancy between the generated text length and the instruction requirement.

**Meaningless**   During the generation of the long text, we observed that longer text often leads to more meaningless content, such as semantic repetition or logically contradictory content, which significantly reduces the overall content quality. As shown in Figure 13, LLM generates redundant and incomprehensible text.

We present more error cases in Table 25 and 26.

# L  FURTHER DISCUSSIONS

## L.1  FUTURE RESEARCH DIRECTIONS

**Long Text Generation Data**    From Table 2 and Table 3, we can observe that LongWriter-GLM4-9B and Suri-I-ORPO can generate significantly longer content compared to other open-source or proprietary LLMs. This is due to their specialized data and alignment algorithms for long text generation tasks. Typically, alignment data follow the paradigms of long-input-short-output or short-input-short-output, with the former mainly aimed at enhancing the LLMs' long-context understanding capabilities. Consequently, it often leads to the model tending to generate short content, resulting in a bias towards shorter outputs. By adding a certain proportion of long text generation data to the alignment data (i.e., long-input-long-output and short-input-long-output), the model can more evenly produce both short and long content, thus preventing the output distribution from being biased towards shorter content to meet different user needs. Therefore, balancing the proportions of different lengths of data is also an area worth exploring. Besides, as we understand, there is a lack of high-quality natural or synthetic long text generation data in the field of LLMs. Constructing high-quality long text generation data is a crucial research direction for future long text generation tasks. We believe that the following approaches can be explored. (1) The first approach involves using the LLM itself to synthesize long text generation data. A simple method is to break down complex instructions into detailed sub-instructions and have the LLM complete each sub-instruction, which is then concatenated. (2) The second approach is the reverse construction of instructions. There are numerous high-quality long texts available on the internet, such as blogs, stories, novels, papers, etc. By constructing instructions for these long texts, a set of high-quality data can be synthesized. It's important for these instructions to be more detailed and include constraints specifically related to long text generation, in order to clearly differentiate them from traditional instructions used for short texts. (3) Regarding natural long text generation data, it is advisable to focus on open-ended instructions, like discussions on specific topics in forums, under legal conditions.

**Inherent Connections in Context Window**    From Table 4, we can observe that models with enhanced long-context understanding capabilities can generate longer content, but the quality of the content tends to degrade. We believe that this phenomenon is related to the model's context window. LLMs are autoregressive models, and during the pre-training stage, the loss is calculated on each token. At this stage, for long-context understanding, each token needs to attend to previous distant tokens. Similarly, for long text generation, distant generated token also needs to attend to previous tokens. From the perspective of the completion, both are akin to dividing a complete text into either short-long or long-short segments. Therefore, we believe there is a correlation between the two in this stage. However, during the alignment stage, the loss is only computed on the response, and the data primarily consists of long-input-short-output or short-input-short-output. This leads to the model's distribution towards shorter content. LLMs with enhanced long-context understanding capabilities typically use more long-text data either during the pre-training or alignment stage. This effectively strengthens the relevance of tokens over a longer range within the context window, enabling the model to comprehend and generate longer texts. Consequently, the length of the generated text increases, but the quality declines, likely due to the insufficient distribution of long text generation data during training. Based on this assumption, we believe that long-context understanding and long text generation capabilities are correlated during the pre-training stage. In the context of current mainstream LLMs with long-context understanding capabilities, the question then arises: can we design an efficient algorithm that uses a small amount of data to activate a model's long text generation capability or shift the model's distribution from generating predominantly short content to producing content with a balanced length? We believe this is a worthwhile and valuable research direction.

**Long Text Generation System**    Whether RecuurentGPT (Zhou et al., 2023) or LongWriter, generating long text generation data involves employing a multi-agent concept to accomplish the task. Given the current model's limitations in generating a large number of words, the multi-agent method divides a specific task into multiple subtasks, each handled by an individual LLM. These subtasks are then summarized to complete the original task. This system or methodology represents an alternative approach to long text generation. We believe there is considerable room for improvement in current methods, making the Long Text Generation System a promising area for future research.

However, it is important to note that HelloBench evaluates the end-to-end long text generation capabilities of LLMs, whereas the Long Text Generation System is not end-to-end.

**Tradeoff Between Quality and Length**   From Table 2, we can observe that there is a tradeoff between the length and the quality of text generated by LLMs. Models like LongWriter-GLM4-9B and Suri-I-ORPO can generate longer content, but the quality of the generated text degrades. Improving both the length and quality of text generated is one of the future research directions.

## L.2   OTHERS

**Timeliness and Ground Truth of HelloBench**   HelloBench does not have ground truth. For HelloBench, we have adhered to the principles of practicality and timeliness when collecting data. We collected real user data to ensure originality and made efforts to collect the latest data available online, minimizing the possibility of data leakage. Moreover, HelloBench focuses on evaluating long text generation rather than having a correct answer or ground truth. Therefore, even if some data leakage might occur, we believe it would not significantly impact the evaluation results for the reason that all the data in HelloBench are open-ended. HelloEval evaluates the quality, factuality, and completeness of the generated text, which differs from standard evaluation. In summary, we believe that HelloBench will not face serious data leakage issues, even as time progresses.

**Customizability of HelloBench**   Many parts of the HelloBench can be customized. For example, the checklists for each subcategory in HelloBench can be customized, and the scores for these checklists can be fitted using another human annotation data or directly assigned by users. Additionally, the prompt wrapping in HelloBench is also replaceable. These features make HelloBench customizable and able to meet the needs of various evaluators.

**Impact of Generation Parameters**   We believe that the generation parameters of LLMs can impact the final evaluation results. However, searching for the optimal generation parameters across different LLMs is time-consuming, labor-intensive, and costly. To ensure a fair comparison of the results from different LLMs, we set the same generation parameters for a relatively fair evaluation. The evaluation results from our LLM-as-a-Judge are fully reproducible, as we set the seed to 42 to ensure reproducibility.

**Length Bias in LLM-as-a-Judge**   In LLM-as-a-Judge, comparing model responses in a pairwise manner can lead to a bias where LLMs tend to prefer longer responses (Dubois et al., 2024). In this work, the evaluation is in a single manner, as we believe it's challenging to evaluate the quality of two lengthy responses, whether for LLMs or humans. By evaluating in a single manner, we do not face the aforementioned length bias. Additionally, our evaluation tasks involve long text generation, so the model is inherently required to produce a longer response. Thus, the preference for longer responses may not necessarily be a bias. In summary, our evaluation method does not have the issue of length bias.

## M   LIMITATIONS

**LLM-as-a-Judge**   Our experiments show that while HelloEval achieves the highest correlation with human evaluation compared to other methods, the correlation is still quite low, around 30. This indicates that evaluation methods based on LLM-as-a-Judge have limitations. However, HelloEval has still achieved relatively better results compared to others. Given the rapid updates and the large number of available LLMs, relying solely on human evaluation would be very time-consuming and labor-intensive, making it impossible to create a comprehensive leaderboard. Therefore, despite its limitations, using LLM-as-a-Judge remains a commonly used evaluation approach at this stage.

**Experiments on more LLMs**   We primarily conduct experiments on mainstream LLMs and lack exploration of other LLMs.

**Multilingualism**   We lack research on multilingualism settings. We will explore the long text generation capabilities in more languages in the future.

## N  SOCIAL IMPACT AND POTENTIAL BIAS

LLMs have been observed to exhibit inherent biases, generating content that may contain discrimination in various aspects such as politics, gender, and race (Das et al., 2024; Ferdaus et al., 2024) due to biased training data. The harmful stereotypes manifested in the generated content can contribute to the oppression of those at social margins (Weidinger et al., 2021). Therefore, in various long text generation fields such as creative writing and story continuation, it is crucial to ensure that the relevant long texts generated by LLMs do not contain harmful stereotypes. Additionally, LLMs are prone to hallucinations, often generating information that is factually incorrect or non-existent (Huang et al., 2023; Sahoo et al., 2024). This issue is particularly prominent in applications requiring high accuracy, such as academic paper editing and news writing, where the dissemination of incorrect information can have serious consequences. Ensuring that LLMs generate reliable and accurate long texts is essential to maintain the credibility of the generated content.

We hope that HelloBench serves as an exemplary platform for future researchers, facilitating the development of reliable and controllable LLM algorithms for long text generation, thereby mitigating societal issues such as the proliferation of fake news or generating content that is discriminatory based on gender or race.

## O  LICENSE OF HELLOBENCH

Figure 14: License of HelloBench.

