# OpenReview forum: "HelloBench: Evaluating Long Text Generation Capabilities of Large Language Models"
_ICLR.cc/2025/Conference — ICLR 2025 Conference Withdrawn Submission_

### Official Review · Reviewer_4N33 · 2024-10-31

**Soundness:** 3
**Presentation:** 3
**Contribution:** 2
**Rating:** 3
**Confidence:** 4

**Summary:**

The paper introduces HelloBench, a benchmark designed to evaluate the long text generation capabilities of LLMs. It categorizes long text generation tasks into open-ended QA, summarization, chat, text completion, and heuristic text generation based on Bloom’s Taxonomy and proposes HelloEval, a human-design evaluation method. The authors conduct experiments on approximately 30 LLMs, revealing limitations in their long text generation capabilities.

**Strengths:**

1. The proposed benchmark is well-categorized and well-grounded.

2. The metric HelloEval is quite innovative and wisely designed. It is also intuitive that using human judgments to induce metric parameters can lead to better human alignment.

3. The experiments are thorough and yield useful findings.

**Weaknesses:**

1. I am not satisfied with the benchmark. While the benchmark is well-categorized, it is not large-scale and does not have good coverage. Its contribution is also very incremental. For example, regarding open-ended questions, the authors only collected 200 samples from a single source Quora. I believe there are existing human-collected benchmarks that are very large-scale. ELI5 is such a representative.

2. To the best of my knowledge, Quora does not allow the crawling of its data. The benchmark may cause policy violations.

3. The analysis of the results lacks depth. While the paper mentions limitations in the models' capabilities, it does not explore the underlying reasons for these limitations or suggest potential improvements.

4. The paper claims that current LLMs struggle with long text generation, but it does not adequately discuss the implications of these findings for the development of future models.

**Questions:**

See weaknesses, plus, does Quora allow data crawling?

**Details Of Ethics Concerns:**

From Quora policy: https://www.quora.com/about/tos

4.4 Restricted Uses. You represent and warrant that you will not:

Access, search or collect data from the Quora Platform (through automated or other means, including artificial intelligence or machine learning) (1) to create derivative works of Our Content and Materials; (2) to train or develop any AI, large language models or machine learning algorithms on Our Content or Materials; (3) to create any service competitive to the Quora Platform; or (4) for other commercial purposes except as expressly permitted by these Terms of Service or the written consent of Quora.

---

> ### Author Response · Authors · 2024-11-24
>
> # Dear Reviewer
> Thank you very much for your valuable comments and questions. We appreciate the time and effort you have put into reviewing our manuscript. Below, we will address your concerns and provide further clarifications.
> ## W1
> HelloBench is sufficiently **large-scale**, has **good coverage**, and makes **significant contributions**.
> 1. **Large-scale**: HelloBench has a total of 647 samples. In comparison, previous long text generation benchmarks like ProxyQA[1] have only 100 samples and LongBench-Write[2] has 120 samples. Besides, considering checklist-wise evaluation costs, we need 647 * 2,500 = 1,617,500 tokens, which equals **$16**. Maintaining an acceptable number of samples is also necessary to control evaluation costs.
> 2. **Good Coverage**: See G.Q1 in General Response.
> 3. **Significant Contributions**: HelloBench makes significant contributions in several aspects. (1) **Data**: It fills the gap in the benchmark for long text generation by addressing issues such as low coverage, limited data volume, and a narrow range of task types found in the previous benchmark. (2) **Evaluation Method**: It includes annotated Human Evaluation data, designs high-quality checklists, applies weighted scores to the checklists, and introduces HelloEval, which enhances accuracy and reduces the cost of subjective evaluation. (3) **Analysis**: we have conducted thorough evaluation experiments, analyzing the weaknesses and error cases of current LLMs from multiple aspects, and providing extensive ablation experiments, offering directions for future research.
> ## W2
> HelloBench has **NOT VIOLATED Quora's Terms of Service**:
> 1. We did not use automated scripts to scrape data from Quora. Instead, we created an account, sorted posts in popular topics by time, and manually collected the data.
> 2. According to Quora's Terms of Service Section 4.4, Quora prohibits data collection and scraping for four purposes: creating derivative works, training AI, creating any competitive service, and commercial use. We adhered to these guidelines. Our use is academic, we did not create derivative works or competitive services, and we did not train AI models using Quora's data. Additionally, based on the Terms of Service's Section 4.5, HelloBench’s data collection process is completely reasonable and legal.
> ## W3
> The experiment and analysis sections of our work includes:
> - **S4.2**: The performance of LLMs in long text generation.
> - **S4.3**: The explicit length constraints.
> - **S4.4**: The relationship between long input and long output.
> - **S4.5**: The effectiveness of HelloEval.
> - **S5.2, Appendix K**: Analyzing the error modes.
>
> Firstly, we have conducted detailed experiments from multiple aspects, analyzing long text generation capabilities of mainstream LLMs. Secondly, after identifying the models' weaknesses, we aim to understand why LLMs performed poorly by analyzing low-scoring LLMs' outputs and summarizing four main error modes. Finally, we provided directions for future work (see line 503 - line 510 and line 2486 - line 2545). Thus, our work includes a complete research trajectory, including problem identification, evaluation dataset construction, experimental evaluation, error analysis, and future research directions. The overall approach and analysis are **comprehensive** and **in-depth**.
> ## W4
> S5.2 and Appendix L discusses the conclusions of our experiments and the research directions that these conclusions suggest (see line 2486 - line 2545). We observed that:
> 1. There is a correlation within the context window, as LLMs are autoregressive models, long input with short output and short input with long output are both related to the model's intrinsic capability to encode long texts.
> 2. Constructing long text generation data can significantly increase the output length of the model. Examples like LongWriter and Suri[3] illustrate this well.
> 3. There is a correlation between the length of the generated text and its quality. As the output length increases, the quality tends to decrease.
> 4. Multi-agent system Methods (like RecurrentGPT[4]) can improve the system's long text generation capability, but these methods are not end-to-end.
>
> As a generative model, LLM faces challenges in long text generation tasks. We believe that research in this field should progress gradually. The current training data, model architecture, and algorithms may not be the most suitable for long text generation tasks. Our work aims to make a significant step forward in this field, guiding subsequent research and promoting the importance of long text generation in the industry, thereby improving the user experience with LLMs.
> ## Q
> See W2, Quora allows data crawling in the condition of HelloBench.
>
> ## Reference
> [1] Proxyqa: An alternative framework for evaluating long-form ...
>
> [2] Longwriter: Unleashing 10,000+ word generation from ...
>
> [3] Suri: Multi-constraint instruction following for ...
>
> [4] RecurrentGPT: Interactive Generation of (Arbitrarily)...

---

> > ### Comment · Reviewer_4N33 · 2024-11-30
> > **Official Comment by Reviewer 4N33**
> >
> > Thanks for your rebuttals. While I appreciate the effort, I am not fully convinced by your responses. I understand the associated costs, but a dataset size of 647 cannot be considered large-scale. While I acknowledge its good coverage, I find it difficult to agree that it constitutes a significant contribution.
> >
> > Regarding data collection, I still have concerns about compliance with Quora’s Terms of Service, specifically Section 4.4, which prohibits data scraping and collection. Although your method may not involve automated crawling, manual collection could still be considered a data collection method. I recommend reaching out to Quora for explicit permission.
> >
> > If you can solve the permission issue with Quora’s policies, I will raise my score to 5.

---

> > > ### Author Response · Authors · 2024-12-02
> > >
> > > Thanks for your comments. We will further Clarify on Quora's policy
> > >
> > > 1. Based on Quora's Terms of Service, section 4.4
> > >
> > > 4.4 Restricted Uses. You represent and warrant that you will not:
> > > Access, search, or collect data from the Quora Platform (through automated or other means, including artificial intelligence or machine learning) (1) to create derivative works of Our Content and Materials; (2) to train or develop any AI, large language models or machine learning algorithms on Our Content or Materials; (3) to create any service competitive to the Quora Platform; or (4) for other commercial purposes except as expressly permitted by these Terms of Service or the written consent of Quora.
> > >
> > > Quora prohibits data collection and scraping for four specific purposes. In collecting data from Quora, HelloBench did not violate these rules. (1) We did not create derivative works of Quora's Content and Materials. (2) We did not train or develop AI models using Quora's data. Specifically, we created an evaluation dataset to assess model capabilities. (3) We did not create any service that competes with the Quora Platform. (4) Our purposes are academic.
> > >
> > > 2. Based on Quora's Terms of Service, section 4.5
> > >
> > > 4.5 Permission to Crawl. If you operate a search engine, subject to the Restricted Uses section above, we conditionally grant permission to crawl the Quora Platform subject to the following rules: (1) you must use a descriptive user agent header; (2) you must follow robots.txt at all times; (3) your access must not adversely affect any aspect of the Quora Platform’s functioning; (4) you must make it clear how to contact you, either in your user agent string, or on your website if you have one.
> > >
> > > Quora allows data collection and scraping under specific conditions compatible with section 4.4, and the following rules must be observed. We adhered to these rules: (1) We did not use automated scripts for scraping; instead, we created accounts to collect data, thus there is a user agent header. (2) We followed the robots.txt requirements and only collected questions from Quora without scraping any other information. (3) We did not affect any aspect of the Quora Platform’s functioning. (4) We included our personal email in the account to make it easy for Quora officials to contact us.
> > >
> > > 3. To ensure the legality of HelloBench, we will include these claims in our paper and project repository. The legality of data collection is crucial, and we will keep track of this aspect. In fact, we have tried to contact Quora officially to further confirm the legality of HelloBench’s data collection process. We reached out through email and by leaving questions on the "Contact Us" section of their website. However, due to time constraints, we have not received a timely response. Once we receive a response, we will adjust or further explain our project content based on the feedback.
> > >
> > >
> > > As the discussion phase will be closed on December 3rd 11:59 pm AoE, we would like to kindly ask you to take a look at our responses and reevaluate our work based on our clarifications. Please let us know whether our response addresses your concerns or whether there is any further detail we can provide to help address these concerns.
> > >
> > > Thank you again for dedicating your time to reviewing our paper.

---

> > > > ### Comment · Reviewer_4N33 · 2024-12-03
> > > > **Official Comment by Reviewer 4N33**
> > > >
> > > > That's where my permission concern is:
> > > >
> > > > ```4.4 Restricted Uses. You represent and warrant that you will not: Access, search, or collect data from the Quora Platform (through automated or other means, including artificial intelligence or machine learning) (1) to create derivative works of Our Content and Materials; (2) to train or develop any AI, large language models or machine learning algorithms on Our Content or Materials;```
> > > >
> > > > You collected data (manually), and this benchmark is used to benchmark and develop AI and large language models in general. This seems a violation. I suggest contacting Quora for explicit permission, my score would be 5.
> > > >
> > > > Note ```4.5 Permission to Crawl``` is not an exception from ```4.4 Restricted Uses``` at all. ```4.5``` means if you do not violate any from ```4.4```, you can crawl but must obey ```4.5```.
> > > >
> > > > My concern about the dataset's contribution remains, so 5 would be my final rating if Quora explicitly said they allowed it.

---

### Official Review · Reviewer_Jq6j · 2024-11-03

**Soundness:** 2
**Presentation:** 4
**Contribution:** 2
**Rating:** 5
**Confidence:** 4

**Summary:**

The paper proposes HelloBench, a benchmark for long text generation inspired by Bloom's taxonomy of cognitive abilities. Furthermore, it proposes HelloEval, a technique to quantitatively evaluate the performance of LLMs initially assessed as checklists. The authors propose training a linear regression model to adjust weights of different checks obtained by human judges to the overall evaluation score, and then use these trained weights when working with an LLM judge.

The authors evaluate a range of recent LLMs on HelloBench, proprietary and open-source, large and small. The evaluation elicited various insights like most LLMs tend to generate output at around 1000 tokens (even those with max_tokens of 16384 or more), text generation quality decreases with response length (especially going outside the usual length of 1000 tokens)

They also conducted a correlation analysis of the HelloEval comparing it to a series of traditional automated text generation metrics and found much stronger Spearman correlation and much lower p-value that all the other methods.

**Strengths:**

* a new long text generation evaluation benchmark HelloBench is proposed which contains multiple tasks inspired by Bloom taxonomy of cognitive abilities
* an evaluation technique is presented for obtaining numerical scores from qualitative checklist-based assessments. It shows superior correlation to human judgements that a series of traditional metrics
* a comprehensive study of a wide range of LLMs is conducted on HelloBench eliciting insights of model's struggles to generate high-quality outputs at higher lengths.

**Weaknesses:**

* the Bloom's taxonomy sounds inspiring but the mapping of different dimensions to text generation tasks looks superficial and containing overlaps. "...we have selected the most
 suitable task for each cognitive level" - this is not obvious to me, needs justification / proof

**Questions:**

N/A

---

> ### Author Response · Authors · 2024-11-24
>
> # Dear Reviewer
> Thank you very much for your valuable comments and questions. We appreciate the time and effort you have put into reviewing our manuscript. Below, We will address your concerns and provide further clarifications.
> ## Weakness
> We will answer it in two parts:
> 1. **Why use Bloom's taxonomy?** Firstly, Bloom's taxonomy is a well-known classification method in the field of education, widely recognized and used by educators. Bloom's taxonomy aligns well with the cognitive levels of LLMs. For HelloBench, Bloom's taxonomy serves as the framework for the entire benchmark. Its main function is to provide researchers with a clear path when constructing the benchmark. Different levels correspond to different sub-tasks, and once sub-tasks corresponding to Bloom's levels are identified, the tasks can then be collected and constructed. A shortfall of previous related work (ProxyQA[1], LongBench-Write[2]) is the incomplete task coverage. With the help of Bloom's taxonomy, we can build a long text generation benchmark with high coverage. Bloom's taxonomy ensures the benchmark is comprehensive and highly covered. Moreover, Bloom's taxonomy minimizes overlap between different data, as each piece of data corresponds to a sub-task, and the overlap between sub-tasks is minimal.
> 2. **Are the sub-tasks aligned appropriately with Bloom's levels?** Bloom's taxonomy in HelloBench serves to (1) define the architecture of the entire benchmark, (2) enhance the coverage and completeness of HelloBench (3) reduce overlap between data. HelloBench includes five sub-tasks, which are the most frequently occurring tasks in real-world scenarios (see G.Q1 in General Response). In our work (see line 133 - line 161), we explain the correspondence between sub-tasks and levels and why they are mapped this way. Thank you for your suggestion; we have revised it to "We have selected representative tasks that frequently appear in real-world scenarios and mapped them to the cognitive level."
>
> Thank you again for your suggestion. We noticed that you pointed out only one weakness and did not raise other questions. If you have additional questions, please feel free to ask, and we will promptly address and respond to them.
>
> ## Reference
> [1] Haochen Tan, Zhijiang Guo, Zhan Shi, Lu Xu, Zhili Liu, Xiaoguang Li, Yasheng Wang, Lifeng Shang, Qun Liu, and Linqi Song. Proxyqa: An alternative framework for evaluating long-form text generation with large language models. arXiv preprint arXiv:2401.15042, 2024.
>
> [2] Yushi Bai, Jiajie Zhang, Xin Lv, Linzhi Zheng, Siqi Zhu, Lei Hou, Yuxiao Dong, Jie Tang, and Juanzi Li. Longwriter: Unleashing 10,000+ word generation from long context llms. arXiv preprint arXiv:2408.07055, 2024.

---

> > ### Author Response · Authors · 2024-12-02
> > **Looking forward to Feedback as Discussion Deadline Approaches**
> >
> > Hi, we sincerely thank you very much for these constructive comments and evaluation of our manuscript. As the discussion phase will be closed on December 3rd 11:59 pm AoE, we would like to kindly ask you to take a look at our responses and reevaluate our work based on our clarifications. Please let us know whether our response addresses your concerns or whether there is any further detail we can provide to help address these concerns.
> >
> > Thank you again for dedicating your time to reviewing our paper.

---

### Official Review · Reviewer_A2vB · 2024-11-04

**Soundness:** 3
**Presentation:** 3
**Contribution:** 3
**Rating:** 6
**Confidence:** 4

**Summary:**

This paper develops a new LLM evaluation benchmark, namely HelloBench, focusing on evaluating LLM’s long text generation capability, filling the missing piece in the current LLM evaluation landscape. HelloBench covers 5 tasks in 38 subcategories, totaling 647 examples, constructed by manual selection from the web and some existing benchmarks. It focuses on open-ended tasks and targets at generation over 1000 words. To evaluate LLMs, the authors further propose HelloEval, a checklist-based LLM-as-a-Judge method that shows positive correlation with human evaluation. Experiments on popular open-source and proprietary LLMs reveal their insufficiency in long text generation.

**Strengths:**

* The benchmark, HelloBench, presents a timely effort on evaluating long text generation for LLM evaluation.
* The evaluation method, HelloEval, provides an automatic way to evaluate LLMs, saving time and effort.
* The experiments are conducted over many popular LLMs, making the findings more reliable and convincing.
* The findings offer insights on the insufficiency of existing LLMs on long text generation capability.

**Weaknesses:**

While HelloEval shows the highest and significant correlation with human evaluation, its overall spearman correlation is just 0.32. It’s not high enough to assume that improvements on HelloEval indicate real gains on long text generation.

**Questions:**

1. What’s the licence of HelloBench? This becomes more important for evaluation benchmarks.
2. Is the correlation analysis for HelloEval based on the annotations from the preparation stage? If so, there may be a risk of overfitting since HelloEval adopts checklist weights derived from this stage?
3. How many annotations did you use for correlation analysis? Is the number large enough to reach a significant conclusion? 0.32 is not a very high correlation value. Please include a plot for HelloEval score and human annotation score: I assume this plot looks more like a cloud rather than a line, so as to remind the others of the risk of HelloEval.
4. In lines 363-364, the authors compared the HelloEval scores between QA/text completion and summarization/chat. Is the scores of different tasks directly comparable?
5. Please provide a detailed section explaining what the HelloEval score means. For example, what does an increase of 1 in HelloEval indicate? What quality gains could be considered as significant? Can we compare scores across sub-tasks?
6. In line 1021, the authors state “By doing so, we guarantee that the data are not leaked for
the reason that they are all test samples and that their quality remains relatively high.” This is not true: test data may already leaked to LLMs.

---

> ### Author Response · Authors · 2024-11-24
>
> # Dear Reviewer
> Thank you very much for your valuable comments and questions. We appreciate the time and effort you have put into reviewing our manuscript. Below, we will address your concerns and provide further clarifications.
> ## Weakness
> Firstly, 0.32 is **not** considered low. In related work, the Spearman correlation in GPTScore[1] ranges from 0.1 to 0.4. In the paper "Is LLM-as-a-Judge Robust? Investigating Universal Adversarial Attacks on Zero-shot LLM Assessment," the Spearman correlation ranges from 0.1 to 0.5. Spearman correlation is meaningful when used comparatively to show differences between evaluation methods. Secondly, in our experimental setup, we compared three LLM-as-a-Judge based evaluation methods and six traditional metrics (lines 448 - 472). Our HelloEval achieved significantly the highest performance. Additionally, many traditional metrics show negative correlations with human evaluations, further proving that traditional metrics are inadequate for evaluating text generation tasks and underscoring the importance and necessity of HelloEval. Lastly, there is also inconsistency among different annotators (see lines 1877 - 1889), making it even more challenging to achieve high correlation values.
> ## Q1
> The license for HelloBench is available in the GitHub repository for this project. Due to the double-blind review process, we have not included the repository link in our paper. Thank you for your suggestion; we have included our license in the appendix of our paper (see line 2612 - line 2634)
> ## Q2
> These two parts do **not** overlap, so there will be no overfitting issue. During the preparation stage, we had five annotators label the human evaluation results of four LLMs, and these results were only used to fit the weighted scores. In the effectiveness experiment of HelloEval, we invited three annotators to label the human evaluation results of seven LLMs. Due to cost reasons, we selected only the summarization task for annotation because only the summarization task has reference answers to compute rouge and bleu metrics. The specific experimental results can be found in Table 23 (see line 2249 - line 2260).
> ## Q3
> Continuing to Question 2, we have a total of 7\*100\*3=2100 annotations. This quantity is large enough to achieve a confident result, and at the same time, by observing the **p-value** in Table 5, which is very low, it further indicates the confidence of the results. For the plot, we have included the relationship between annotator one's human evaluation scores with HelloEval scores in the appendix of our paper (see line 2222- line 2245).
> ## Q4
> The scores of different subtasks are **comparable**. Firstly, the score ranges of different subtasks are consistent. Secondly, although the checklists for different tasks vary, they generally follow similar evaluation aspects (instruction following, comprehensiveness, easy-to-understand, factuality, overall, see Appendix E).
> ## Q5
> The scores obtained through HelloEval are not significantly different from the normal scores. HelloEval focuses on aligning the scores with human evaluation. However, due to rescaling, the benefit curve of the rescaled scores is different from the original 0-100 scale, where each point of improvement represents 1/400 of the overall scale. In terms of positive or negative scores, if the rescaled score is positive, the original score exceeds 75, which means that the scores for each checklist item are generally above 0.75, indicating that your response quality is above the acceptable line. Based on the response to Q4, scores of different subtasks are comparable.
> ## Q6
> The main intent of this statement in the paper is to highlight that these data are less likely to leak compared to other data. Although some LLMs might have been passively trained on these test data, at least it ensures that they are not actively trained on them, as LLMs are not trained specifically toward the test data. However, we agree that this statement could be more precise, and we have revised it to: "By doing so, we guarantee that the probability of data leakage is minimized, and the quality of these data is ensured." Thank you for your suggestion.
> ## Reference
> [1] Jinlan Fu, See-Kiong Ng, Zhengbao Jiang, and Pengfei Liu. Gptscore: Evaluate as you desire. arXiv preprint arXiv:2302.04166, 2023.

---

> > ### Comment · Reviewer_A2vB · 2024-12-02
> >
> > Thanks for your response! I'd like to keep my current judgement.

---

> ### Author Response · Authors · 2024-12-02
>
> Thank you for your feedback! If you have any other questions or issues, please feel free to let me know.

---

### Official Review · Reviewer_bkWt · 2024-11-05

**Soundness:** 3
**Presentation:** 3
**Contribution:** 3
**Rating:** 5
**Confidence:** 4

**Summary:**

This paper presents HelloBench, a comprehensive benchmark designed to assess the long text generation capabilities of large language models (LLMs). Additionally, the authors introduce HelloEval, an automatic evaluation method that leverages LLMs-as-a-Judge to efficiently evaluate checklist results associated with each long text generation task. Through extensive experimentation across approximately 30 mainstream LLMs, the work reveals significant limitations in the long text generation capabilities of these models, including an inability to generate text exceeding 4000 words.

**Strengths:**

HelloBench encompasses a diverse array of long text generation tasks, such as open-ended QA and summarization, thereby offering a holistic evaluation framework for assessing LLMs' long text generation capabilities. The proposed HelloEval methodology reduces the time and labor associated with human evaluation, while maintaining a strong correlation with human judgments. The authors conducted experiments across 30 mainstream LLMs, providing valuable insights into the current limitations of long text generation.

**Weaknesses:**

1) Omission of Prior Work: The paper fails to adequately acknowledge and compare its methodology with ProxyQA [1], a pioneering framework specifically designed for evaluating long-form text generation capabilities of LLMs. Both methodologies assess generated content indirectly through evaluators to ensure adherence to specific standards. However, ProxyQA employs a query-specific checklist known as proxy questions, while HelloBench uses a more general checklist, which can not adaptively provide query- and semantic-aware checklists. The motivations and insights of both approaches appear to align closely.

2) To provide valuable context, it would be beneficial to include a comparison with ProxyQA in Table 1. Additionally, an analysis of the correlation between the results of ProxyQA and those of the proposed HelloBench would significantly strengthen this paper. If this analysis demonstrates that HelloBench aligns well with ProxyQA while offering more challenging and representative tasks, it would enhance the credibility of the work.

3) It appears that all the key components of HelloBench or HelloEval, such as the six levels of Bloom's Taxonomy, the concept of LLM-as-a-Judge, the checklist-based evaluation method, and the dataset collection approach, have already been proposed in existing works. Furthermore, the ProxyQA work has already investigated the long-text generation benchmark and evaluation method, which overlaps with the focus of this study. The novelty and contribution of the work are quite limited.

[1] Tan, H., Guo, Z., Shi, Z., Xu, L., Liu, Z., Feng, Y., ... & Song, L. ACL 2024. ProxyQA: An alternative framework for evaluating long-form text generation with large language models.

**Questions:**

Lack of Robustness Analysis: Beyond examining the correlation between Hellobench and human evaluation, it would be beneficial to conduct win rate analysis or similar experiments (CI test). This would help determine if the proposed method consistently produces reliable and firm judgments, as even top-performing LLMs or human evaluators can generate inconsistent evaluation results.

---

> ### Author Response · Authors · 2024-11-24
>
> # Dear Reviewer
> Thank you very much for your valuable comments and questions. We appreciate the time and effort you have put into reviewing our manuscript. Below, we will address your concerns and provide further clarifications.
> ## Weakness 1
> We have noticed the impressive work of ProxyQA[1] and have referenced it in our paper (see line 521 - line 529). However, we did not include it in the comparison table (Table 1). For now, we have updated and supplemented this in our work (see line 089 - line 096). Regarding your question "which cannot adaptively provide query- and semantic-aware checklists," the checklists in HelloBench are very granular. We have designed specific checklists for each subcategory. Our dataset includes a total of 38 subcategories, so on average, each set of checklists corresponds to around 20 samples.
> ## Weakness 2
> Yes, we have added a comparison with ProxyQA in Table 1. In Section 4.5, we compared the performance of HelloEval with several methods (MTBench[2], WildBench[3], etc.) under the same settings (see line 448 - line 472), and HelloEval achieved the best results. When conducting experiments, we also considered whether to compare more evaluation methods, such as ProxyQA. However, these methods are generally coupled with specific datasets. For instance, the combination of proxy questions and meta questions in ProxyQA cannot be applied to HelloBench without annotating a set of proxy questions specifically for HelloBench. We also wanted to compare the correlation of ProxyQA and HelloEval under the same settings, but we found that ProxyQA’s proxy questions are not open-sourced. Additionally, since the evaluated LLMs are not consistent, we cannot compare the ranking consistency between ProxyQA and HelloEval. If you have any good suggestions, we are happy to compare the correlation between ProxyQA and HelloEval.
> ## Weakness 3
> Contributions of HelloBench:
> 1. **Evaluation Method**: We propose HelloEval, an evaluation method aligned with human evaluation. This method can be plugged into traditional subjective evaluation methods to enhance their alignment with human evaluation, reduce the cost of human evaluation, and improve evaluation efficiency.
> 2. **Benchmark HelloBench**: Different from ProxyQA, which focuses on a single task, HelloBench covers a wider range of tasks, and it aims at real user scenarios. HelloBench contains 647 entries, is large-scale, and has undergone multiple steps (cleaning, deduplication, filtering) to ensure high data quality (see line 1223 - line 1259).
> 3. **Experimental Analysis**: We have conducted detailed experiments, analyzing the long text generation capabilities of current LLMs from various aspects (implicit word limit, explicit word limit, the relationship between long text comprehension, and long text generation capabilities). Furthermore, after identifying the models' weaknesses, we explore why the models perform poorly by analyzing low-scoring samples and summarizing four main error modes. Finally, we suggest future research directions (see line 503 - line 509 and line 2486 - line 2545). As such, our work presents a complete research pathway, including problem identification, dataset construction, experimental evaluation, error analysis, and suggestions for future work.
>
> Both ProxyQA and HelloBench aim to bring more attention to the field of long text generation and promote improvements in LLMs' long text generation capabilities. Therefore, we believe that both ProxyQA and HelloBench have made significant contributions.
> ## Question
> See G.Q2 in General Response
> ## Reference
> [1] Tan, H., Guo, Z., Shi, Z., Xu, L., Liu, Z., Feng, Y., ... & Song, L. ACL 2024. ProxyQA: An alternative framework for evaluating long-form text generation with large language models.
>
> [2] Lianmin Zheng, Wei-Lin Chiang, Ying Sheng, Siyuan Zhuang, Zhanghao Wu, Yonghao Zhuang, Zi Lin, Zhuohan Li, Dacheng Li, Eric Xing, et al. Judging llm-as-a-judge with mt-bench and chatbot arena. Advances in Neural Information Processing Systems, 36, 2024.
>
> [3] Bill Yuchen Lin, Yuntian Deng, Khyathi Chandu, Faeze Brahman, Abhilasha Ravichander, Valentina Pyatkin, Nouha Dziri, Ronan Le Bras, and Yejin Choi. Wildbench: Benchmarking llms with challenging tasks from real users in the wild. arXiv preprint arXiv:2406.04770, 2024.

---

> > ### Comment · Reviewer_bkWt · 2024-11-29
> > **Official Comment by Reviewer bkWt**
> >
> > Thank you for your efforts in preparing the rebuttal. I have reviewed your response and have decided to maintain my current score. My concerns remain regarding the contribution of this work, which appears to be incremental, and the coverage or scope of the evaluation data is limited.

---

> ### Author Response · Authors · 2024-12-02
>
> Thank you for your feedback!  If you have any other questions or issues, please feel free to let me know.

---

### Author Response · Authors · 2024-11-24
**General Response**

Thanks a lot for handling/reviewing our submitted manuscript. We would like to thank the reviewers for their thoughtful and constructive comments and suggestions. By addressing each of the issues raised by the reviewers, we believe that the quality and clarity of our HelloBench can be improved a lot. The general responses are summarized as follows:

## The Revised Manuscript
Considering the constructive suggestions given by reviewers, we will provide a new version of our manuscript before the deadline of the author response period (November 27 at 11:59pm AoE).

## G.Q1: The coverage of HelloBench?
HelloBench has a high coverage rate and closely aligns with real user scenarios. To illustrate this point, we list the statistics we obtained from WildChat (see line 1049 - line 1118). The 15 most common long text generation task types in WildChat[1] are: report writing (HTG), guide generation (HTG), science problem solving (Chat), academic writing (HTG), content continuation (TC), question answering (OEQA), rewriting (TC), script writing (HTG), creative writing (HTG), idea generation (HTG), explanation (HTG), data analysis (Summ), character creation (HTG), curriculum development (Chat), and question generation (HTG). It can be observed that all 15 categories are covered by HelloBench's sub-tasks, achieving a **100%** coverage rate. In comparison, the previous benchmark such as ProxyQA only covered QA tasks, and LongBench-Write only covered creative writing tasks.
## G.Q2: The win-rate experiments of HelloEval.
Regarding the win-rate experiment mentioned in Reviewer bkWt.Question, we believe there are two ways to conduct it. The first method is to compare all models pairwise, but this approach is costly and may have the issues of length-bias, as analyzed in our paper (see line 2569 - line 2576). The second method is to compare all models against a single, relatively strong model (e.g., GPT-4o), which has a lower evaluation cost than the first method. Here, we used the second approach for our supplementary experiment. We had three annotators compare the win-rate of different models against GPT-4o, using checklists as the evaluation standard (see line 1728 - line 1772). Due to time constraints, we sampled 100 samples each and tested 5 LLMs. The results of the supplementary experiment are as follows:

| Model | winrate |
| --- | --- |
| Claude-3.5-Sonnet |  47.33 |
| Mistral-Lage-API | 55.67 |
| Gemma-2-27B | 47.00 |
| LLaMA-3.1-70B | 37.33 |
| Gemini-1.5-Pro | 45.33 |

We can observe that the results of the evaluated models show strong consistency in ranking with the main experiment. The main experiment (Mistral > Claude > Gemini > Gemma > LLaMA) and the supplementary experiment (Mistral > Claude > Gemma > Gemini > LLaMA), further prove the effectiveness of HelloEval. Thank you again for your suggestion. We have included this part in the appendix (see line 2261 - line 2279).
## Reference
[1] Wenting Zhao, Xiang Ren, Jack Hessel, Claire Cardie, Yejin Choi, and Yuntian Deng. Wildchat: 1m chatgpt interaction logs in the wild. arXiv preprint arXiv:2405.01470, 2024.

---

### Author Response · Authors · 2024-11-24
**Summarization on the responses**

Thanks for handling/reviewing our submitted manuscript: "HelloBench: Evaluating Long Text Generation Capabilities of Large Language Models". We would like to thank the reviewers for their insightful and constructive comments and suggestions. By addressing each of the issues raised by the reviewers, we believe that the quality and clarity of our HelloBench can be improved a lot. The major responses are summarized as follows:

(1). We have carefully discussed the contributions of HelloBench and the difference between HelloBench and previous works (like ProxyQA) (See Reviewer bkWt. W1&W2&W3 and Reviewer 4N33. W1).

(2). We have additionally conducted the win-rate experiments and further demonstrated the effectiveness of HelloEval (See Reviewer bkWt.Q).

(3). We have discussed the Spearman Correlation of HelloEval and detailed the experimental settings of the effectiveness experiments of HelloEval (See Reviewer A2vB.W&Q2&Q3).

(4). We have provided more clarification on the license and data collection permission of HelloBench (See Reviewer A2vB.Q1, Reviewer 4N33. W2&Q).

(5). We have provided more clarification on the HelloEval scores and the sub-tasks of HelloBench is comparable (See Reviewer A2vB.Q4&Q5).

(6). We have discussed the reasons behind the choice of Bloom's Taxonomy (See Reviewer Jq6j.Weakness).

(7). We have provided more clarification on the potential improvements of long text generation capabilities and the research directions provided by our work (See Reviewer 4N33.W3&W4).

Again, we would like to sincerely thank you very much for these constructive comments and evaluation for our manuscript.

---

### Author Response · Authors · 2024-11-27
**Revised Paper Submitted and Looking forward to feedback**

## Dear Reviewers
Considering the constructive suggestions given by reviewers, we have provided a new version of our manuscript and appended the old version of our manuscript in the supplementary material. We made the following changes:

1. Considering the suggestion given by Reviewer bkWt. W1&W2, we have appended the comparison between HelloBench and ProxyQA in Table 1 (see line 089 - line 096).
2. Considering the suggestion given by Reviewer A2vB. Q1, we have appended the LICENSE of HelloBench in Appendix O (see line 2612 - line 2634).
3. Considering the suggestion given by Reviewer A2vB. Q3, we have appended the plot of HelloEval Scores with Human Evaluation Scores in Figure 12  (see line 2221 - line 2245).
4. Considering the suggestions given by Reviewer A2vB. Q6 and Reviewer Jq6j.W, we have revised some of the sentences in our paper.
5. Considering the suggestion given by Reviewer 4N33.W3, we have detailed our future research directions in Appendix L (see line 2486 - line 2545).
6. Considering the suggestion given by Reviewer bkWt.Q, we have supplemented win-rate experiments and appended in the Appendix J.3 (see line 2261 - line 2279).

Thanks again for the thoughtful and constructive comments and suggestions, by addressing these issues raised by the reviewers, we believe that the quality and clarity of our HelloBench can be improved a lot.

**Lastly, we would like to kindly ask you to take a look at our responses and revised paper and reevaluate our work based on our clarifications. Please let us know whether our response addresses your concerns or whether there is any further detail we can provide to help address them. We appreciate your time and consideration!**

---

### Note · Authors · 2024-12-12

I have read and agree with the venue's withdrawal policy on behalf of myself and my co-authors.